# Exploring Connections Between Memorization And Membership Inference

## Abstract

Membership Inference Attacks (MIAs) aim to identify specific data samples within the private training dataset of machine learning models. Many practical black-box MIAs require query access to the data distribution to train *shadow models*. Prior literature presents bounds for the adversary's success by making connections to overfitting (and its connections to differential privacy), noting that overfit models with high generalization error are more susceptible to attacks. However, overfitting does not fully account for privacy risks in models that generalize well. We take a complementary approach: by observing that label memorization can be reduced to membership inference, we are able to present theoretical scenarios where the adversary will always successfully (i.e., with extremely high advantage) launch an MIA. We proceed to show that these attacks can be launched at a fraction of the cost of state-of-the-art attacks. We confirm our theoretical arguments with comprehensive experiments; by utilizing samples with high memorization scores, the adversary can (a) significantly improve its efficacy regardless of the MIA used, and (b) reduce the number of shadow models by nearly two orders of magnitude compared to state-of-the-art approaches.

## 1 Introduction

Machine learning (ML) models are often trained on large volumes of data that is custom curated for the task at hand. Collecting and processing this data is both tedious and time-consuming, and requires significant investment. Additionally, the data used to train ML models is oftentimes sensitive, and protecting its privacy is paramount (Choquette-Choo et al., 2021). For instance, ML models trained for medical applications involve health-related information pertaining to patients; the privacy of this data is protected by regulations such as HIPAA (Annas, 2003).

The widely held belief was that, since the data is processed in non-intuitive ways to obtain an ML model, purely interacting with the model would not leak information about the data used to train it. However, *membership inference* (MI) aims to infer if a particular record was present (or absent) in the training data given access to a trained model. Follow-up research aims to determine the *advantage* of such an MI adversary *i.e.,* how likely is the adversary to succeed in their attempts, mostly based on connections to differential privacy (Mahloujifar et al., 2022; Sablayrolles et al., 2019; Yeom et al., 2018). In this paper, we focus on the fact that such MI advantage (*i.e.,* adversary success) widely varies across different data samples, and study how the adversary can identify "more vulnerable" data to employ more successful and less expensive MI attacks (MIAs).

Our analysis leverages the label memorization framework (Feldman, 2020; Brown et al., 2021; Feldman & Zhang, 2020). Recall that overfitting (colloquially) occurs when a model performs well on training data but poorly on unseen data, indicating a lack of generalization. However, a model can generalize well and still memorize specific training examples, especially outliers or unique data points. This memorization poses privacy risks, as adversaries can exploit it to extract sensitive information from the model. Therefore, focusing on label memorization allows for a more accurate assessment of privacy vulnerabilities, as it directly addresses the unintended retention of specific training data, irrespective of the model's generalization performance (Feldman, 2020).

Our main contribution here is to view the MIA framework through the lens of memorization, and provide the *first formal bound of MI advantage as a function of label memorization.* This bound is particularly interesting as it unearths the characteristics of samples for which MIAs are most successful, as label memorization is a data-dependent phenomenon. While it may seem intuitive that highly memorized samples are easier to attack, this relationship has not been formally or generally characterized in prior work. In fact, recent studies (Carlini et al., 2022a) show that even non-memorized test points can be misclassified as members, indicating that the connection between memorization and membership inference vulnerability is far from trivial. This motivates our effort to move beyond intuition and rigorously quantify this relationship using tools from information theory and hypothesis testing.

It is also worth noting that many MIAs necessitate the training of a substantial number of *shadow models*, thereby creating a computational bottleneck (Shokri et al., 2017; Song & Mittal, 2021; Watson et al., 2021; Ye et al., 2022); we establish a connection between label memorization and the computational efficiency of MIAs. By framing MIA as a hypothesis testing problem, akin to previous works (Carlini et al., 2022a; Ye et al., 2022), we note that the sample complexity of hypothesis testing corresponds to the number of shadow models. This allows us to *provide a formal bound on the sample complexity in terms of label memorization*, implying that MIAs targeting highly memorized samples necessitate fewer shadow models to achieve successful results.

Lastly, we conduct extensive evaluation over multiple vision datasets and models to support our argument that our explanation of disparate impact yields both (a) more successful, and (b) computationally inexpensive MIAs compared to state-of-the-art approaches. These experiments serve as support for our theoretical formulation, and are not meant to be immediately practical. With highly memorized data, we are able to achieve AUROC of 0.99 and TPR of 100% at 0.1% FPR with less than 20 shadow models, whereas achieving the same success with random data often requires more than 200 shadow models.

**Related Work:** While previous works have focused on bounding MIA success using overfitting and differential privacy (Yeom et al., 2018; Ye et al., 2022), memorization is a more ubiquitous phenomenon than overfitting, as demonstrated in several papers (den Burg & Williams, 2021; Rocks & Mehta, 2022; Tirumala et al., 2022), and can be more precisely characterized (Recht, 2024). Carlini et al. (2022a) proposed LiRA, an effective attack to distinguish whether a sample was used for training or not, whose formulation is essentially identical to subsampling-based memorization approximation (as we highlight in Appendix C.2). Their follow-up work (Carlini et al., 2022b) notes that memorization is a relative concept, showing experimentally that removing highly likely members (as detected by LiRA) eliminates the most memorized samples, revealing a new layer of memorized data. A variant of this phenomenon is also shown by Jagielski et al. (2022). However, none of these works formally establish a theoretical connection between memorization and MIA success; to our knowledge, we make the first effort. For an extended discussion of related work, see Section 5.

## 2    Background and Problem Overview

In this section, we introduce the background and formalism that we use throughout the paper.

### 2.1    Memorization

Distribution $\mathcal{D}$ captures the space of inputs and outputs, from which a dataset $S$ of size $s$ is sampled (*i.e.,* $S \sim \mathcal{D}^s$) such that $S = \{z_i\}_{i=1}^s$. Each $z_i$ is of the form of input-label pair $(x_i, y_i)$, where $x_i \in \mathcal{X}$ is the space of inputs (*e.g.,* images) and $y_i \in \mathcal{Y}$ is the space of outputs (*e.g.,* labels). Using this dataset, a learning algorithm $L$ (*e.g.,* stochastic gradient descent (Bottou, 2012)) can create a trained model *i.e.,* $\theta \sim L(S)$ by minimizing a suitable objective $\mathcal{L}$ (*e.g.,* cross-entropy loss (Zhang & Sabuncu, 2018)).

**Label Memorization:** ML models are vastly overparameterized, and are known to fit to even random labels (Feldman, 2020). This is termed *memorization*. For a sample $z_i$, given dataset $S$ (without $z_i$), label memorization is formalized as the absolute difference in correctness probabilities [1]:

$$\texttt{mem}(L, S, z_i) := \left| \Pr_{\theta \sim L(S^{(i)})}[\theta(x_i) = y_i] - \Pr_{\theta \sim L(S)}[\theta(x_i) = y_i] \right| = \Pr_{\theta \sim L(S^{(i)})}[\theta(x_i) = y_i] - \Pr_{\theta \sim L(S)}[\theta(x_i) = y_i] \quad (1)$$

---

[1]For most practical algorithms, we expect the value in Equation 1 to be non-negative; see Section 4.1 in Feldman (2020).

where $S^{(i)} = S \cup \{z_i\}$. This is a point-wise difference in marginal prediction behavior for the fixed pair $z_i$. This definition differs from sequence-level or reconstruction-style memorization (e.g., Brown et al., 2021), which focus on verbatim reproduction of training records.

## 2.2 Membership Inference

**Security Game:** The MI game defines the interaction between two parties: an adversary $\mathcal{A}$ aiming to perform MI, and a challenger $\mathcal{C}$ that is responsible for training. We assume $\mathcal{A}$ knows the data distribution $\mathcal{D}$ and the learning algorithm $L$ but not the randomness used by $L$. The output of $L$ belongs to the set $\Theta$.

1. $\mathcal{C}$ picks a dataset $S$ sampled according to $\mathcal{D}$.
2. $\mathcal{A}$ picks $z \sim \mathcal{D}$ and sends it to $\mathcal{C}$.
3. $\mathcal{C}$ picks a random bit $b_\mathcal{C} \leftarrow \{0, 1\}$. If $b_\mathcal{C} = 0$, let $S' = S$; otherwise $S' = S \cup \{z\}$. $\mathcal{C}$ executes $L$ on $S'$ and sends the result $\theta_L \in \Theta$ to $\mathcal{A}$.
4. $\mathcal{A}$ guesses a bit $b_\mathcal{A}$. If $b_\mathcal{C} = b_\mathcal{A}$, then the output of the game is 1 (indicating that the adversary won the game); otherwise the output of the game is 0 (indicating that the adversary lost the game).

The advantage of the adversary is written as the difference between $\mathcal{A}$'s true and false positive rates as follows (Yeom et al., 2018),

$$\mathrm{Adv}(L, \mathcal{A}) \;=\; \Pr(b_\mathcal{A} = 1 \mid b_\mathcal{C} = 1) - \Pr(b_\mathcal{A} = 1 \mid b_\mathcal{C} = 0) \tag{2}$$

Let $C_\mathcal{A}$ be a class of adversaries (*e.g.,* probabilistic polynomial time algorithm adversaries). We can define $\mathrm{Adv}(L, C_\mathcal{A})$ as $\sup_{\mathcal{A} \in C_\mathcal{A}} \mathrm{Adv}(L, \mathcal{A})$. When the class $C_\mathcal{A}$ is implicit from the context we still write it as $\mathrm{Adv}(L, \mathcal{A})$. The random variable corresponding to the *MI* game is written as $MI_{L,\mathcal{A}}$.

**Threat Model:** We define $MI_{L,\mathcal{A}}^s$ separately to denote the *stronger game* where $\mathcal{A}$ also has access to the actual dataset $S$ not just the distribution $\mathcal{D}$ (Mahloujifar et al., 2022) (*i.e.,* $\mathcal{A}$ picks $z \in S$ in step 2). This aligns with the "neighboring dataset" formulation in differential privacy, where the adversary's goal is to distinguish between $S$ and $S \cup \{z\}$. Such a setting enables a clean, information-theoretic analysis of per-sample privacy leakage and serves as a worst-case baseline: if an algorithm is robust here, it will remain robust under weaker adversaries. Observe that such an assumption is not far-fetched. In scenarios where trusted auditors verify data deletion through MIAs (Ma et al., 2022), the auditors (in this case, the adversary) are assumed to know both the dataset $S$ and the point being deleted $z$.

**Important To Note:** In this model, the adversary aims to identify, with high success, the membership of specific points. This does not significantly deviate from the setting of existing work. More so, our game definition is a variant of definition 3.3 in (Ye et al., 2022), where the adversary chooses the sample to include in the game. Privacy analysis under such a stronger definition (i.e., assuming a worst-case adversary) is useful, as the failure of the adversary would imply the robustness of the learning algorithm to privacy attacks. A useful analogy is the IND-CPA game in cryptography (Wikipedia contributors, 2025), where security is analyzed in a setting that favors the adversary, ensuring that defenses remain effective even under strong attack assumptions.

Also note that prior works (Carlini et al., 2022a; Zarifzadeh et al., 2024) argue that an effective MIA is the one that *reliably* identifies few members of a sensitive dataset. We follow the same setup[2]. Our privacy adversary wants to evaluate their MIA's (formalized as algorithm $\mathcal{A}$) true-positive rate (TPR) at low false-positive rates (FPR); success implies high TPR at low FPR regimes.

## 2.3 Hypothesis Testing

A hypothesis test is a method of statistical inference where one checks if the data at hand is sufficient to support a particular hypothesis. Let $P_0$ and $P_1$ denote two distributions with support $\mathcal{X}$. A hypothesis test

---

[2]Throughout this work, "shadow models" refer to models trained on subsamples or approximations of $S$, consistent with the original usage in Shokri et al. (2017). This definition encompasses both cases where shadow models are trained on independent draws from $\mathcal{D}$ and cases where they are trained on subsets of the actual $S$, as in many practical MIAs (e.g., Carlini et al., 2022a).

$T : \mathcal{X}^{\star} \to \{0, 1\}$ takes a sequence of $n$ elements $\sigma = \{o_1, \cdots, o_n\}$ and predicts 0 (*i.e.,* $\sigma$ is generated by $P_0$) or 1 (*i.e.,* $\sigma$ is generated by $P_1$). There are two types of errors associated:

- **Type I error**: The probability that the test outputs $P_1$ if $P_0$ is true *i.e.,* $\Pr_{\sigma \sim P_0^n}(T(\sigma) = 1)$.
- **Type II error**: The probability that the test outputs $P_0$ if $P_1$ is true *i.e.,* $\Pr_{\sigma \sim P_1^n}(T(\sigma) = 0)$.

The ideal goal of a hypothesis test is to achieve Type I error below an application specific threshold.

The advantage of a test $T$ is defined as,

$$Adv_n(T) = \Pr_{\sigma \sim P_0^n}(T(\sigma) = 0) - \Pr_{\sigma \sim P_1^n}(T(\sigma) = 0) \tag{3}$$

**Distance between two distributions**: For two probability distributions $P_0, P_1$ over the same domain, we use $TV(P_0, P_1)$ to denote the total variation distance between them, which is the largest possible difference in probabilities that the two distributions can assign to the same event. We also define $H^2(P_0, P_1)$ as the square of the Hellinger distance between $P_0$ and $P_1$. The TV distance and Hellinger distance are related by the following well-known inequalities: $H^2(P_0, P_1) \leq TV(P_0, P_1)$.

**Log-Likelihood Ratio (LLR) Test:** The Neyman-Pearson lemma (Neyman & Pearson, 1933) states that the LLR test achieves the best Type II error for a given bound on the Type I error, and is optimal.

$$\text{LLR}(\sigma) = \log \frac{P_0^n(\sigma)}{P_1^n(\sigma)} = \log \prod_{i=1}^{n} \frac{P_0(o_i)}{P_1(o_i)} \tag{4}$$

If $\text{LLR}(\sigma) \geq \kappa$ (for some constant $\kappa$), the test outputs $P_0$, and otherwise outputs $P_1$. Consider the soft version of the above LLR test, $\text{sLLR}(x)$, which outputs 0 with probability $g(x)$ and 1 with probability $1 - g(x)$, where $g(x)$ is:

$$\frac{e(P_0, P_1)}{1 + e(P_0, P_1)}$$

where $e(P_0, P_1)$ denotes $\exp(\frac{1}{2} \log \frac{P_0(o)}{P_1(o)})$. Canonne et al. (2019) describe the advantage of the sLLR test using a single sample as follows:

**Lemma 1** *For any two distributions $P_0, P_1$, the advantage of sLLR test with $n = 1$ is $Adv_1(sLLR) = H^2(P_0, P_1)$.*

**Sample Complexity:**

*Sample complexity $M_\alpha^{P_0, P_1}(T)$ of a test $T$ is defined such that for all $n \geq M_\alpha^{P_0, P_1}(T)$, the advantage $Adv_n(T) \geq \alpha$. The sample complexity of the optimal test $T^\star$ serves as the lower bound for the number of samples necessary to achieve an error probability $1 - \alpha$:*

$$M_\alpha^{P_0, P1}(T^\star) = \min_T \ M_\alpha^{P_0, P1}(T)$$

where $T^\star$ corresponds to LLR test in Equation 4 by Neyman-Pearson lemma.

**Lemma 2** *The sample complexity of the optimal test $T^\star$ to distinguish $P_0$ from $P_1$ is $\Theta(\frac{1}{H^2(P_0, P_1)})$.*

This bound is tight (*i.e.,* both lower-bounded and upper-bounded) (Bar-Yossef, 2002; Canonne et al., 2019).

## 2.4 Problem Statement

We wish to establish a theoretical connection between game $MI_{L,\mathcal{A}}^s$ and label memorization. By doing so, we will demonstrate that utilizing highly memorized data samples can significantly improve both success and computational efficiency (§ 3) of adversaries (c.f., § 4).

We study a setting where the MI adversary can strategize its game by choosing samples that are highly likely to be memorized (*i.e.,* a large value in Equation 1) to add in game $MI^s_{L,\mathcal{A}}$. Here, we assume the existence of a memorization oracle $\mathcal{O}_{mem}$ that returns an approximate memorization score for any sample chosen by the adversary. These samples may come from the same distribution $\mathcal{D}$ that was used to obtain $S$, or can come from a different distribution (refer § 4 of Carlini *et al.* (Carlini et al., 2021)). In our experiments in § 4, data points $z$ are sampled from a different distribution than $S$, and the memorization oracle is instantiated by running the algorithm proposed by Feldman and Zhang (Feldman & Zhang, 2020) or TRAK (Park et al., 2023). The presence (or absence) of such points leads to a characteristic signal that the adversary exploits to launch MIAs.

**Our Main Thesis:** We wish to prove that assuming $\mathcal{A}$ has complete knowledge of the dataset $S$ used to train a machine learning model using algorithm $L$, for a sample $z^*$,

- $\mathcal{A}$ can successfully[3] identify its membership in $S$ if $\mathtt{mem}(L, S, z^*) = 1$ *i.e.,* $\mathrm{Adv}(L, \mathcal{A}) = 1$.

- If $\mathcal{A}$ utilizes $m$ shadow models for launching an MIA for a sample $z$, it requires $m^* \leq m$ shadow models if $\mathtt{mem}(L, S, z^*) = 1$.

## 3 Memorization Bounds Membership Inference

We aim to establish a formal understanding of the connection between memorization and membership inference. We present a theoretical analysis that addresses two key aspects: (a) the memorization score of a point serves as a lower bound for the MI adversary's success (in determining membership of that point), and (b) higher values of memorization score correlate with improved computational efficiency of MIAs. To achieve these conclusions, we view the MI game through the lens of hypothesis testing. Our work aligns with recent endeavors to connect MIAs and hypothesis testing (Ye et al., 2022; Carlini et al., 2022a), and we provide a rigorous description of this connection, exploring various assumptions regarding the adversary's capabilities.

### 3.1 Memorization and MIA Efficacy

Let $f : \Theta \to \mathbb{R}^\ell$ (*e.g.,* $f$ might extract the logits corresponding to the pre-softmax layer of a network (Carlini et al., 2022a)). The adversary $\mathcal{A}$ may not have direct access to the actual dataset $S \sim \mathcal{D}$ constructed by the challenger (as in the default security game in § 2.2). However, leveraging the learning algorithm $L$, the adversary can initiate a random process involving sampling as follows: sample a new dataset $\hat{S} \sim \mathcal{D}$ and can either,

1. Execute $L$ on $\hat{S}$ to obtain $\theta_L \in \Theta$. Then output $f(\theta_L)$, which is characterized by distribution $P_0$. Note that $P_0$ on $\mathbb{R}^\ell$ depends on $\mathcal{D}$ and also the randomness of $L$.

2. Execute $L$ on $\hat{S} \cup \{z\}$ to obtain $\theta_L \in \Theta$. Then output $f(\theta_L)$, which is characterized by distribution $P_1$.

**Problem to Solve:** $\mathcal{A}$ receives a new $\theta_L \in \Theta$ and needs to decide if $f(\theta_L) \in \mathbb{R}^\ell$ was generated by distribution $P_0$ or $P_1$. In other words, $\mathcal{A}$ needs a hypothesis test $T_{\mathcal{D},L}$ that, given *one* sample $o \in \mathbb{R}^\ell$, outputs 0 or 1 corresponding to distributions $P_0$ and $P_1$. We define the following terms:

$$\mathcal{T}_1 = \Pr_{o \sim P_0} (T_{\mathcal{D},L} = 0),$$

$$\mathcal{T}_2 = \Pr_{o \sim P_1} (T_{\mathcal{D},L} = 0)$$

*i.e.,* the adversary in this MI game launches the hypothesis testing with $T_{\mathcal{D},L}$, and this gives us the following bound: $Adv^T(L, \mathcal{A}) = \mathcal{T}_1 - \mathcal{T}_2$ where $Adv^T(L, \mathcal{A})$ is $Adv_1(T_{\mathcal{D},L})$ (refer Equation 3). Next, we outline the specific hypothesis tests that the adversary can employ to launch effective MIAs in two scenarios.

---

[3]For success defined in § 2.2.

**Scenario 1: $\mathcal{A}$ perfectly knows $P_0$ and $P_1$.** Assume that the adversary knows the density function for $P_0$ and $P_1$. Given $\theta_L$, $\mathcal{A}$ can use the sLLR test to output 0 and 1. Then we immediately get the following theorem by Lemma 1.

> **Theorem 1** $Adv(L, \mathcal{A}) \geq H^2(P_0, P_1)$ *where $\mathcal{A}$ uses sLLR test as described above.*

**Scenario 2: $\mathcal{A}$ does not know $P_0$ and $P_1$.** Consider the case when the adversary only knows $P_0$ and $P_1$ implicitly, and does not know their probability density functions. Then $\mathcal{A}$ can make a parametric assumption on $P_0$ and $P_1$; for instance, assume that they are two $l$-dimensional multivariate normal distributions $N(\mu_1, \Gamma_1)$ and $N(\mu_2, \Gamma_2)$ [4]. $\mathcal{A}$ decides the output based on the following strategy:

1. Empirically estimate the parameters of $P_0$ and $P_1$. Generate $m$ datasets $\hat{S}_1, \cdots, \hat{S}_m$ sampled from $\mathcal{D}$ (recall that $\mathcal{A}$ knows $\mathcal{D}$). Execute $L$ on $\hat{S}_1, \cdots, \hat{S}_m$ and $\hat{S}_1 \cup \{z\}, \cdots, \hat{S}_m \cup \{z\}$ and generate two sets: $Z_0 = \{f(\theta_1), \cdots, f(\theta_m)\}$ and $Z_1 = \{f(\theta_{1,z}), \cdots, f(\theta_{m,z})\}$. Then, estimate the corresponding parameters *i.e.,* $(\mu_0, \Gamma_0)$ for $Z_0$. and $(\mu_1, \Gamma_1)$ for set $Z_1$. Note that $m$ corresponds to the number of shadow models in the literature (Shokri et al., 2017; Sablayrolles et al., 2019; Carlini et al., 2022a). $Z_0$ represents the **out** case where the data $z$ is excluded during training, and $Z_1$ represents the **in** case where $z$ is a member of the training dataset.

2. Decide whether $f(\theta_L)$ given by the challenger is in $P_0$ and $P_1$ using $\text{sLLR}(f(\theta_L))$ test with normal distributions parameterized by $N(\mu_0, \Gamma_0)$ and $N(\mu_1, \Gamma_1)$.

The Hellinger distance $H^2(P_0, P_1)$ where $P_0 \sim N(\mu_0, \Gamma_0)$ and $P_1 \sim N(\mu_1, \Gamma_1)$ is given by:

$$1 - \frac{(\det(\Gamma_0)\det(\Gamma_1))^{\frac{1}{4}}}{\det(\frac{\Gamma_0 + \Gamma_1}{2})^{\frac{1}{2}}} \times \zeta$$

where $\zeta = \exp\left\{-\frac{1}{8}(\mu_0 - \mu_1)^\top (\frac{\Gamma_0 + \Gamma_1}{2})^{-1}(\mu_0 - \mu_1)\right\}$ (where $\top$ denotes transpose). This immediately becomes the lower bound for MI advantage by Theorem 1.

**Note:** Recall that there is a one-sided version of the problem as well: $\mathcal{A}$ receives $\theta_L \in \Theta$ and needs to decide if $f(\theta_L)$ was generated by distribution $P_0$ or not. In other words, the second distribution $P_1$ is absent.

**Under Stronger Adversary:** Assume that $\mathcal{A}$ has access to not only data distribution $\mathcal{D}$ and learning algorithm $L$ but also knows the exact dataset $S$. Symmetrically, the MI advantage in Equation 2 can be rewritten as

$$\begin{aligned}
\text{Adv}(L, \mathcal{A}) &= \Pr(b_{\mathcal{A}} = 1 \mid b_{\mathcal{C}} = 1) - \Pr(b_{\mathcal{A}} = 1 \mid b_{\mathcal{C}} = 0) \\
&\geq \Pr(\mathcal{A}(L(S \cup \{z\})) = 1) - \Pr(\mathcal{A}(L(S)) = 1)
\end{aligned} \tag{5}$$

where the probability is over the randomness in $L$ and $\mathcal{A}$.

In the above expression, $\mathcal{A}$ is the shorthand for the algorithm used by the adversary to predict the bit $b_A$. It can be viewed as a randomized function from $\Theta \times \mathcal{X}$ to $\{0, 1\}$. Let $r_L$ and $r_{\mathcal{A}}$ be the random strings corresponding to $L$ and $\mathcal{A}$. If $r_L$ and $r_{\mathcal{A}}$ are independent, we can rewrite Equation 5 as

$$\Pr_{\theta \sim L(S \cup \{z\})} \alpha(\mathcal{A}, \theta, z) - \Pr_{\theta \sim L(S)} \alpha(\mathcal{A}, \theta, z) \tag{6}$$

where $\alpha(\mathcal{A}, \theta) = \Pr_{r_{\mathcal{A}}}(\mathcal{A}(\theta, z) = 1)$. Note that $\alpha(\mathcal{A}, \theta, z)$ is a *deterministic* function from $\Theta \times \mathcal{X}$ to $\{0, 1\}$. This is a standard argument in analyzing games in cryptography; w.l.o.g, we can consider deterministic adversaries.

---

[4]Such parametric approximation has been used in the literature (Carlini et al., 2022a). Note that other than multivariate normal distribution, any parametric distribution (*e.g.,* exponential) can be assumed for $P_0$ and $P_1$. The only requirement would be that the parameters of the distribution should be efficiently estimated, and preferably, the Hellinger distance should be expressed in a concise, closed form.

Now, let us consider the following adversary: $\mathcal{A}(\theta, z) = \mathbb{1}_{\theta(x)=y}$, where $z = (x, y)$; $x$ is data and $y$ is the corresponding label. In this case, Equation 6 simplifies to:

$$Adv^m(L, \mathcal{A}) = \Pr_{\theta \in L(S \cup \{z\})}(\theta(x) = y) - \Pr_{\theta \in L(S)}(\theta(x) = y)$$
$$= \text{mem}(L, S, z)$$

which immediately proves that:

---

**Theorem 2** $Adv(L, \mathcal{A}) \geq \text{mem}(L, S, z)$.
*This implies that if the adversary can choose $z$ with a high memorization score [a], the MI advantage can be increased.*

---

[a]Caveat: in practical setup, we assume the existence of memorization oracle $\mathcal{O}_{\text{mem}}$ that gives the approximate memorization score given a sample. We leave the realization of a tractable, and effective $\mathcal{O}_{\text{mem}}$ as an important future work.

---

Recall that for the advantage $\text{Adv}(L, \mathcal{A})$ we implicitly take the supremum over adversaries in a certain class. Hence, an advantage $\beta$ with a specific adversary becomes a lower bound on the advantage of the adversary (see Appendix B.2 for advantage with other various instantiations of the adversary).

### 3.2 Memorization and MIA Efficiency

Finally, based on our analysis in the above two subsections, we provide the connection between the memorization and the sample efficiency of MI game.

Observe that Scenario 2 in § 3.1 is the one which is most general *i.e.,* the adversary knows nothing about both distributions. In such settings, note that the adversaries need to train $m$ shadow models to obtain observations (or "samples") which can be used to approximate distributions, and then hypothesis testing is performed using these approximated distributions. Recall that Lemma 2 (refer § 2.3) defines the optimal sample complexity for any hypothesis test based on the Hellinger distance between two distributions. *Thus, one can observe that the number of shadow models corresponds to the sample complexity of the hypothesis test.* Two natural questions emerge: (a) how does one estimate this sample complexity when the distributions are unknown?, and (b) how many samples do we need to effectively estimate the distribution for hypothesis testing?[5] To answer these questions, we first need to establish a formal relationship between the Hellinger distance, $H^2(P_0, P_1)$, and our memorization score, $\text{mem}(L, S, z)$. However, a direct comparison is challenging: $H^2$ is a global measure of divergence between the entire model distributions, while $\text{mem}$ is a local, point-wise measure. We bridge this gap by making an explicit assumption about how these distributions diverge. Under our *prediction-aligned divergence assumption* (see Appendix B.1 for a detailed proof), we posit that the primary difference between distributions $P_0$ and $P_1$ is captured by the change in the model's prediction on the specific point $z$. This allows us to relate the global $H^2$ distance to the local $\text{mem}$ score via the TV distance, yielding the following key inequality,

$$H^2(P_0, P_1) \leq \text{mem}(L, S, z) \tag{7}$$

By plugging Equation 7 into Lemma 2, we get

**Corollary 1** $M_\alpha^{P_0, P_1}(T^\star) = \Omega\left(\frac{1}{\text{mem}(L, S, z)}\right)$ [6].

The implication is that if $\mathcal{A}$ can launch the attack using $z$ with high memorization score, the required number of shadow models can be reduced, which is the major computational bottleneck in most popular attacks (Song & Mittal, 2021; Sablayrolles et al., 2019; Shokri et al., 2017; Carlini et al., 2022a).

---

[5]Both questions have the same implication.
[6]While empirical results suggest this bound is tight in practice, a theoretical upper bound remains an open direction. Nonetheless, we note that this lower bound still meaningfully proves the fundamental relationship between $\text{mem}$ and the minimum cost of an attack represented in terms of sample complexity.

# 4    Experiments

Having established the theoretical connections between memorization and MIA efficacy and efficiency, we now validate them empirically. Our goal here is to test the implications of our theoretical results under controlled conditions, not to design or propose a practical attack pipeline. We assume the existence of an oracle $\mathcal{O}_{\mathrm{mem}}$ that provides memorization scores for arbitrary samples; in practice such scores would need to be approximated, and may be noisy. This assumption allows us to directly evaluate how sample choice, informed by memorization, affects MIA success and efficiency. We stress that these experiments are not meant to serve as "practical" MIAs, but provide more insight on how an adversary may be more successful. We carry out experiments to answer the following questions:

**RQ1.** Are MIAs more performant on samples with high memorization scores compared to randomly chosen OOD samples or samples that belong to under-represented subpopulations (§ 4.2)?

**RQ2.** If an adversary had access to even approximate memorization information, how much could this reduce the computational cost of MIAs (§ 4.3)?

Based on our experiments, we observe that:

**A1:** MIAs that utilize highly memorized data are significantly more effective than those based on other phenomenon *i.e.,* memorization can accurately explain the disparate performance of MIAs; the attack of Carlini *et al.* (Carlini et al., 2022a) with highly memorized data consistently achieves an AUROC of 1.0 across datasets. Moreover, it consistently exceeds TPR of 96% (at 0.1% FPR) and, in some cases, even attains a perfect TPR of 100% (Table 1).

**A2:** An adversary that utilizes data with high memorization scores can achieve a significant reduction in MIA overhead (*i.e.,* training fewer shadow models). Specifically, across all considered dataset choices, the adversary achieves an equivalent level of attack performance with mere 5 shadow models to match the performance with 2000 shadow models under the same setup, a $400\times$ reduction. The same effect is observed even when the adversary utilizes random OOD data (Figure 1), albeit not the same magnitude of reduction.

## 4.1    Setup

We briefly describe our experimental setup used to validate the theory (see Appendix C for further details).

**Datasets.** We consider a training data $D^{\mathrm{tr}}$ that comprises a mixture of two distinct groups (say $D_{\mathrm{O}}^{\mathrm{tr}}$ and $D_{\mathrm{U}}^{\mathrm{tr}}$) with a mixing ratio $\alpha \in [0,1]$ (*i.e.,* to obtain $D^{\mathrm{tr}} = \alpha \cdot D_{\mathrm{O}}^{\mathrm{tr}} + (1-\alpha) \cdot D_{\mathrm{U}}^{\mathrm{tr}}$). In such settings, one dataset is an *over-represented group* (*i.e.,* $D_{\mathrm{O}}^{\mathrm{tr}}$ when the mixing coefficient is greater than 0.5), and the other is an *under-represented group* (*i.e.,* $D_{\mathrm{U}}^{\mathrm{tr}}$). Such composite datasets have been recognized in many contexts; for instance, in federated learning, data in many settings, data is pooled from different sources such as devices from various geographic regions (Li et al., 2020); modern datasets are long-tailed and composed of varied subpopulations (Zhu et al., 2014; Van Horn & Perona, 2017; Babbar & Schölkopf, 2019). We set $D_{\mathrm{O}}^{\mathrm{tr}}$ as either the MNIST train dataset (consisting of 60,000 samples) (LeCun et al., 2010) or the CIFAR-10 train dataset (consisting of 50,000 samples) (Krizhevsky, 2009). We explore three different approaches to define $D_{\mathrm{U}}^{\mathrm{tr}}$:

(a) (**Random OOD**) When $D_{\mathrm{O}}^{\mathrm{tr}}$ is the MNIST train set, $D_{\mathrm{U}}^{\mathrm{tr}}$ is 1000 MNIST samples augmented by approaches proposed by Hendrycks *et al.* (Hendrycks et al., 2022), 1000 SVHN samples, or 6000 CIFAR-10 samples. When $D_{\mathrm{O}}^{\mathrm{tr}}$ is CIFAR-10, then $D_{\mathrm{U}}^{\mathrm{tr}}$ is 1000 samples randomly drawn from the CIFAR-100 train set (Krizhevsky, 2009), or 1000 random CIFAR-10 train data augmented as earlier (Hendrycks et al., 2022). Labeling across both groups was consistent. We ensure that the accuracy (both test and train) of both the under- and over-represented groups is reasonable.

(b) (**Subpopulations**) The adversary identifies random OOD samples that belong to a particular subpopulation (Jagielski et al., 2021): for each $D^{\mathrm{tr}}$ setup as specified in (a), we first preprocess $D^{\mathrm{tr}}$ by extracting the feature representations from the penultimate layer of a DNN trained on $D^{\mathrm{tr}}$ (*e.g.,*

EfficientNet (Tan & Le, 2019) in our case). Then, we apply a PCA projection to further reduce the feature dimensionality to 10. Finally, the KMeans clustering algorithm (Hartigan & Wong, 1979) is applied to cluster the projected $D^{\mathrm{tr}}$ into five subgroups. We identify those samples from $D_{\mathrm{U}}^{\mathrm{tr}}$ belonging to the smallest cluster as the desired subpopulation [7].

(c) (**Singletons**) The adversary has access to an oracle $\mathcal{O}_{\mathrm{mem}}$ and identifies OOD samples, particularly with high memorization scores. We consider two ways of instantiating $\mathcal{O}_{\mathrm{mem}}$; the empirical approximation of self-influence (Feldman & Zhang, 2020), and TRAK (Park et al., 2023) (a more efficient approximation algorithm of self-influence). For each choice of $D_{\mathrm{O}}^{\mathrm{tr}} \bigcup D_{\mathrm{U}}^{\mathrm{tr}}$ as specified in (a), we define *singletons* as data belonging to $D_{\mathrm{U}}^{\mathrm{tr}}$ whose memorization degree exceeds a threshold (0.8 when $D_{\mathrm{O}}^{\mathrm{tr}}$ is MNIST, and 0.5 when $D_{\mathrm{O}}^{\mathrm{tr}}$ is CIFAR-10 [8]).

We choose these three categories of constructing under-represented groups as they represent popular theories for explaining the disparate performance of MIAs on samples. Evaluation on these categories will help better understand which theory is more accurate in explaining the behavior. Note that while these datasets may seem artificially mixed, our comparison is not the absolute success of any single attack, but relative gains for each attack between different types of vulnerable samples (to truly understand the best underlying cause).

**Models.** We follow the setup in Carlini et al. (2022a) and employ two CNN models with varying convolutional filter sizes set to 32 and 64 and two ResNet models (ResNet-9 and ResNet-18) (He et al., 2016). Our primary focus is on the vision setting, as the notion of label memorization is well-studied and evaluated in this setting.

**MIAs.** We consider five representative attacks from the MIA literature (Yeom et al., 2018; Shokri et al., 2017; Song & Mittal, 2021; Sablayrolles et al., 2019; Carlini et al., 2022a). Yeom et al. (2018) employ a very simple attack by thresholding the loss output without necessitating the adversary's access to the training dataset or distribution. On the other hand, other MIAs are built upon the assumption that the MI adversary has query access to the training data distribution for shadow model training (Shokri et al., 2017; Song & Mittal, 2021; Sablayrolles et al., 2019; Carlini et al., 2022a). For those, we train 2000 shadow models for each setting.

**Metrics.** Based on the premise established by Carlini et al. (2022a), for each attack, we report the TPR when a decision threshold $\tau$ is chosen to maximize TPR at a target FPR (*e.g.,* as low as 0.1% in our setting) along with AUROC values (*i.e.,* the probability that a positive example would have a higher value than a negative one). We report the performance when the adversary launches an MIA with random samples in $D^{\mathrm{tr}}$ ("All") vs. selected samples in $D_{\mathrm{U}}^{\mathrm{tr}}$ ("Under-rep.").

### 4.2 Increased MIA Success

Table 1 provides a comprehensive summary of the results obtained from evaluating MIAs across various data settings (see Table 4 in Appendix D.1 for more results).

For every attack, the first row corresponds to the performance of the attacks in the conventional setting, where $D^{\mathrm{tr}}$ is constructed using a single dataset. Subsequently, the next group of three rows represent scenarios where the adversary constructs the training dataset as a mixture of two groups, with variations in the selection of the under-represented group, as outlined in § 4.1. For each setting, we provide the attack success rate under two conditions: (a) when random data is used as the challenge data (denoted as column "All"), as commonly used in literature, and (b) when the adversary specifically utilizes data from the under-represented group (denoted as column "Under-represented") [9]. This comparative analysis allows for a deeper understanding of MIA effectiveness by examining the impact of using data from the under-represented group.

---

[7]We note that hyperparameter choices, such as dimensionality of projection space, number of clusters, etc., are left to the adversary's specifications depending on the setup, and one could always come up with other methodologies to instantiate the subgroup identification.

[8]To ensure an adequate representation of singletons in conjunction with the over-represented population, we choose the threshold with which $|D_{\mathrm{U}}^{\mathrm{tr}}|$ is not significantly smaller than $|D_{\mathrm{O}}^{\mathrm{tr}}|$

[9]When dataset consists of single population (*e.g.,* MNIST, or CIFAR-10), "Under-represented" reports the attack performance on the data samples with memorization scores above 0.4 for MNIST, and 0.8 for CIFAR-10; the smaller threshold is chosen for MNIST since the dataset is less long-tailed. This is to observe if the adversary can still leverage memorization to launch a better attack even in a single dataset setup.

**Takeaway 1:** We observe that every attack *always performs better* with respect to the sample from the under-represented group (*i.e.,* OOD samples, OOD subpopulation, or OOD singletons) than with a random sample (most likely to be drawn from the over-represented groups); given MNIST+augMNIST (OOD), the attack proposed by Shokri *et al.* (Shokri et al., 2017) yields an overall AUROC of 0.51 and TPR of 0.20% under "All". However, we see an increase in the AUROC to 0.72 and TPR to 0.70% when the attack is evaluated on under-represented augMNIST samples (under "Under-represented" of MNIST+augMNIST (OOD) setting). *The most substantial improvement is observed when utilizing OOD singletons in all scenarios*; in the case of MNIST+SVHN (Self-infl.), the attack devised by Sablayrolles *et al.* (Sablayrolles et al., 2019) demonstrates enhanced efficacy from an AUROC of 0.51 and TPR of 2.90% (see "All" of MNIST+SVHN (Self-infl.)) to an AUROC of 1.0 and TPR of 97.55% (see "Under-represented" of MNIST+SVHN (Self-infl.)). This surpasses the attack's performance when evaluated using OOD samples, where the AUROC is 0.81 and TPR is 2.86% (see "Under-represented" of MNIST+SVHN (OOD)), or even using OOD subpopulation, where the AUROC is 0.95 and TPR is 26.0% (see "Under-represented" of MNIST+SVHN (subpop.)). Observe that this phenomenon is true even in scenarios where the dataset is not artificially mixed, *i.e.,* the rows with only MNIST or CIFAR-10.

While we see increased efficiency with singletons in most of the cases, we note that utilizing samples from OOD subpopulations leads to better attack success in some cases; *e.g.,* see the results of Yeom *et al.* (Yeom et al., 2018) with CIFAR-10+CIFAR-100 dataset. One possible explanation for this phenomena could be made by observing the distribution of memorization scores of samples in CIFAR-10+CIFAR-100 dataset (see Fig. 7 in Appendix D.2). We observe that the distribution of memorization scores for the original CIFAR-10 dataset is long-tailed (as in Fig. 7a), but when mixed with a random subset of CIFAR-100 dataset, all samples fall into the lower memorization regime under the score of 0.6 (as in Fig. 7b). That is, even if the adversary constructs the under-represented group using the threshold 0.5, the identified samples are not really highly memorized, and eventually, not any better than being selected by subpopulation identification.

**Takeaway 2:** Nonetheless, it is worth noting that in all scenarios, the attack proposed by Carlini *et al.* (Carlini et al., 2022a) consistently outperforms other attacks. This observation is consistent with the findings reported in their original paper. Moreover, their attack achieves the best efficacy with singletons, even when singleton identification cannot be made reliably (as in the case of CIFAR10 + CIFAR100 dataset). Notably, by utilizing singletons, the attack achieves near-perfect AUROC and TPR even at a low FPR of 0.1%. We believe that the remarkable performance of this attack can be attributed to the inherent connection between the attack itself and the estimation of memorization (further discussion in Appendix E).

### 4.3 Decreased Computational Overhead

In § 3, we theoretically demonstrated that for a sample with a large memorization score, the number of shadow models required to distinguish the **in** case (*i.e.,* when the sample is a member) from the **out** case (*i.e.,* when the sample is not a member) is low. In this section, we provide empirical evidence that this relation holds in practical settings.

In Fig. 1, we plot the success of the MIA by Carlini et al. (2022a) against the number of shadow models needed to achieve it. We use this attack as it demonstrates the most success (refer to results in § 4.2). Specifically, Fig. 1a and Fig. 1b represent scenarios where the model is trained partly using OOD data *i.e.,* $D_U^{tr}$ comprises of random OOD data. Similarly, Fig. 8 and Fig. 1d illustrate cases where the model is trained partly using data from highly memorized OOD data. For the comparison with the cases where the attack is employed using data randomly chosen without consideration of memorization, see Fig. 5 in Appendix D.1.

**Takeaway 1:** Observe that for high memorization score samples, there is a substantial reduction in the number of shadow models needed to achieve a particular success threshold. For example, in Fig. 1d, it is phenomenal that we could achieve AUROC of 1.0 with just one shadow model for **in** case and **out** case each, without necessitating thousands of shadow models.

**Takeaway 2:** We also see that the same phenomenon holds even when the adversary leverages random OOD data. We observe that less than 50 shadow models are sufficient to achieve the same level of AUROC as 2000 shadow models (see Fig. 1a and Fig. 1b), a 40× reduction. These empirical findings suggest that even in situations where the adversary does not have access to a reliable memorization oracle, there is still

| Method | Dataset ($D^{tr}$) | AUROC ↑ All | AUROC ↑ Under-rep. | TPR @ 0.1% FPR ↑ All | TPR @ 0.1% FPR ↑ Under-rep. | Dataset ($D^{tr}$) | AUROC ↑ All | AUROC ↑ Under-rep. | TPR @ 0.1% FPR ↑ All | TPR @ 0.1% FPR ↑ Under-rep. |
|---|---|---|---|---|---|---|---|---|---|---|
| | MNIST | 0.50 | 0.50 | 0.0% | 0.0% | CIFAR-10 | 0.62 | 0.73 | 0.0% | 0.0% |
| | MNIST+augMNIST (OOD) | 0.50 | 0.61 | 0.0% | 0.0% | CIFAR-10+augCIFAR-10 (OOD) | 0.57 | 0.69 | 0.0% | 0.0% |
| | MNIST+augMNIST (Subpop.) | 0.50 | 0.80 | 0.0% | 0.0% | CIFAR-10+augCIFAR-10 (Subpop.) | 0.57 | 0.69 | 0.0% | 0.0% |
| Yeom et al. (2018) | MNIST+augMNIST (Self-infl.) | 0.50 | 0.82 | 0.0% | 0.0% | CIFAR-10+augCIFAR-10 (self-infl.) | 0.57 | 0.71 | 0.07% | 0.59% |
| | MNIST+augMNIST (TRAK) | 0.50 | **0.86** | 0.0% | **3.01%** | CIFAR-10+augCIFAR-10 (TRAK) | 0.57 | **0.73** | 0.10% | **6.16%** |
| | MNIST+SVHN (OOD) | 0.50 | 0.77 | 0.0% | 0.0% | CIFAR-10+CIFAR-100 (OOD) | 0.57 | 0.79 | 0.0% | 0.0% |
| | MNIST+SVHN (Subpop.) | 0.50 | 0.80 | 0.0% | 0.0% | CIFAR-10+CIFAR-100 (Subpop.) | 0.62 | **0.85** | 0.0% | **17.32%** |
| | MNIST+SVHN (Self-infl.) | 0.50 | **0.93** | 0.0% | **2.86%** | CIFAR-10+CIFAR-100 (self-infl.) | 0.63 | 0.76 | 0.0% | 0.0% |
| | MNIST+SVHN (TRAK) | 0.50 | 0.90 | 0.0% | 0.03% | CIFAR-10+CIFAR-100 (TRAK) | 0.63 | 0.76 | 0.0% | 0.0% |
| | MNIST | 0.51 | 0.66 | 0.07% | 3.33% | CIFAR-10 | 0.69 | 0.99 | 0.41% | 53.70% |
| | MNIST+augMNIST (OOD) | 0.51 | 0.72 | 0.20% | 0.70% | CIFAR-10+augCIFAR-10 (OOD) | 0.63 | 0.78 | 0.16% | 1.17% |
| | MNIST+augMNIST (Subpop.) | 0.51 | 0.66 | 0.12% | 3.25% | CIFAR-10+augCIFAR-10 (Subpop.) | 0.60 | 0.78 | 0.12% | 0.76% |
| Shokri et al. (2017) | MNIST+augMNIST (Self-infl.) | 0.51 | **0.99** | 0.15% | **18.88%** | CIFAR-10+augCIFAR-10 (self-infl.) | 0.63 | **0.97** | 0.20% | **39.29%** |
| | MNIST+augMNIST (TRAK) | 0.51 | 0.95 | 0.12% | 15.73% | CIFAR-10+augCIFAR-10 (TRAK) | 0.63 | 0.93 | 0.22% | 38.36% |
| | MNIST+SVHN (OOD) | 0.52 | 0.76 | 0.25% | 0.30% | CIFAR-10+CIFAR-100 (OOD) | 0.69 | 0.85 | 0.21% | 0.28% |
| | MNIST+SVHN (Subpop.) | 0.51 | 0.91 | 0.10% | 1.41% | CIFAR-10+CIFAR-100 (Subpop.) | 0.68 | 0.95 | 0.24% | 20.16% |
| | MNIST+SVHN (Self-infl.) | 0.51 | **0.99** | 0.23% | **72.86%** | CIFAR-10+CIFAR-100 (self-infl.) | 0.70 | **1.0** | 0.45% | **100%** |
| | MNIST+SVHN (TRAK) | 0.51 | 0.99 | 0.24% | 59.47% | CIFAR-10+CIFAR-100 (TRAK) | 0.71 | 0.95 | 0.26% | 14.07% |
| | MNIST | 0.51 | 0.75 | 0.12% | 15.09% | CIFAR-10 | 0.71 | 1.0 | 6.18% | 68.17% |
| | MNIST+augMNIST (OOD) | 0.51 | 0.85 | 0.22% | 0.61% | CIFAR-10+augCIFAR-10 (OOD) | 0.68 | 0.87 | 3.20% | 9.17% |
| | MNIST+augMNIST (Subpop.) | 0.52 | 0.93 | 0.11% | 27.91% | CIFAR-10+augCIFAR-10 (Subpop.) | 0.67 | 0.87 | 3.17% | 6.03% |
| Sablayrolles et al. (2019) | MNIST+augMNIST (Self-infl.) | 0.52 | **0.99** | 0.33% | **46.10%** | CIFAR-10+augCIFAR-10 (self-infl.) | 0.72 | **1.0** | 3.14% | **46.97%** |
| | MNIST+augMNIST (TRAK) | 0.51 | 0.99 | 0.29% | 42.08% | CIFAR-10+augCIFAR-10 (TRAK) | 0.70 | 0.93 | 3.17% | 37.91% |
| | MNIST+SVHN (OOD) | 0.50 | 0.81 | 0.21% | 2.86% | CIFAR-10+CIFAR-100 (OOD) | 0.74 | 0.90 | 5.97% | 37.05% |
| | MNIST+SVHN (Subpop.) | 0.51 | 0.95 | 0.14% | 26.0% | CIFAR-10+CIFAR-100 (Subpop.) | 0.75 | 1.0 | 5.01% | 56.90% |
| | MNIST+SVHN (Self-infl.) | 0.51 | **1.0** | 2.90% | **97.55%** | CIFAR-10+CIFAR-100 (self-infl.) | 0.75 | **1.0** | 6.21% | **82.31%** |
| | MNIST+SVHN (TRAK) | 0.51 | 1.0 | 1.17% | 94.27% | CIFAR-10+CIFAR-100 (TRAK) | 0.75 | 1.0 | 6.01% | 75.68% |
| | MNIST | 0.51 | 0.68 | 0.14% | 2.96% | CIFAR-10 | 0.62 | 0.95 | 3.18% | 19.52% |
| | MNIST+augMNIST (OOD) | 0.51 | 0.70 | 0.16% | 0.53% | CIFAR-10+augCIFAR-10 (OOD) | 0.55 | 0.65 | 1.18% | 5.40% |
| | MNIST+augMNIST (Subpop.) | 0.50 | 0.92 | 0.11% | 11.57% | CIFAR-10+augCIFAR-10 (Subpop.) | 0.54 | 0.62 | 1.01% | 3.20% |
| Song & Mittal (2021) | MNIST+augMNIST (Self-infl.) | 0.51 | 0.98 | 0.10% | 16.12% | CIFAR-10+augCIFAR-10 (self-infl.) | 0.55 | **0.81** | 3.15% | **18.20%** |
| | MNIST+augMNIST (TRAK) | 0.51 | **0.99** | 0.11% | **18.27%** | CIFAR-10+augCIFAR-10 (TRAK) | 0.55 | 0.76 | 2.90% | 16.17% |
| | MNIST+SVHN (OOD) | 0.50 | 0.77 | 0.17% | 2.86% | CIFAR-10+CIFAR-100 (OOD) | 0.60 | 0.89 | 3.01% | 11.25% |
| | MNIST+SVHN (Subpop.) | 0.51 | 0.91 | 0.14% | 9.18% | CIFAR-10+CIFAR-100 (Subpop.) | 0.60 | 0.92 | 2.89% | 15.08% |
| | MNIST+SVHN (Self-infl.) | 0.51 | **0.98** | 2.04% | **14.11%** | CIFAR-10+CIFAR-100 (self-infl.) | 0.61 | **0.98** | 3.13% | **21.11%** |
| | MNIST+SVHN (TRAK) | 0.51 | 0.95 | 2.35% | 11.98% | CIFAR-10+CIFAR-100 (TRAK) | 0.61 | 0.98 | 3.05% | 20.05% |
| | MNIST | 0.52 | 0.89 | 0.88% | 45.98% | CIFAR-10 | 0.87 | 1.0 | 15.25% | 95.47% |
| | MNIST+augMNIST (OOD) | 0.53 | 0.87 | 1.45% | 27.55% | CIFAR-10+augCIFAR-10 (OOD) | 0.73 | 0.93 | 5.96% | 36.73% |
| | MNIST+augMNIST (Subpop.) | 0.52 | 0.96 | 0.33% | 68.29% | CIFAR-10+augCIFAR-10 (Subpop.) | 0.72 | 0.93 | 2.81% | 29.55% |
| Carlini et al. (2022a) | MNIST+augMNIST (Self-infl.) | 0.53 | **1.0** | 1.07% | **100%** | CIFAR-10+augCIFAR-10 (self-infl.) | 0.74 | **1.0** | 6.91% | **98.01%** |
| | MNIST+augMNIST (TRAK) | 0.53 | 0.99 | 1.13% | 93.10% | CIFAR-10+augCIFAR-10 (TRAK) | 0.72 | 0.99 | 6.78% | 88.36% |
| | MNIST+SVHN (OOD) | 0.53 | 0.95 | 2.15% | 51.5% | CIFAR-10+CIFAR-100 (OOD) | 0.87 | 0.96 | 14.62% | 54.42% |
| | MNIST+SVHN (Subpop.) | 0.52 | 0.99 | 0.50% | 78.71% | CIFAR-10+CIFAR-100 (Subpop.) | 0.85 | 0.99 | 12.21% | 89.76% |
| | MNIST+SVHN (Self-infl.) | 0.53 | **1.0** | 1.24% | **100%** | CIFAR-10+CIFAR-100 (self-infl.) | 0.86 | **1.0** | 15.79% | **100%** |
| | MNIST+SVHN (TRAK) | 0.53 | 1.0 | 2.19% | 99.26% | CIFAR-10+CIFAR-100 (TRAK) | 0.87 | 1.0 | 17.89% | 95.56% |

Table 1: **Comparison of representative MIAs on various dataset constructions.** For each attack method, we consider different dataset mixtures for both MNIST and CIFAR datasets. For each choice of MIA and mixture dataset, compare AUROC and TPR between (a) "All" vs. Under-represented (Under-rep.), and (b) OOD vs. Subpopulation (Subpop.) vs. Singletons identified using self-influence (Self-infl.) or TRAK. Singleton selection relies on an *oracle* that provides memorization scores—an idealized assumption used to estimate the *maximum* potential benefit of memorization-guided MIAs. Best results are **boldfaced**.

potential to reduce the computational overhead of MIAs, although to a lesser extent than when utilizing highly memorized samples.

## 5 Related Work

### 5.1 Membership Inference

For many applications, ML models are trained on large corpora of data, some of which are sensitive and private (Shokri et al., 2017). MI is a class of attacks on the privacy of the data used to train the model, and are training distribution-aware (Yeom et al., 2018; Shokri et al., 2017; Carlini et al., 2022a; Jayaraman et al., 2020; Song & Mittal, 2021; Murakonda et al., 2021; Ye et al., 2022; Sablayrolles et al., 2019; Leino & Fredrikson, 2020). Adversaries aim to infer if a sample was present in the training dataset, given access to a trained model. They often do so by checking for effects due to the presence or absence of the point under consideration on a large set of shadow models – those that are trained on varying data subsets to estimate these effects. MIAs were used to test if data is successfully deleted from ML models (Bourtoule et al., 2021), with some caveats (Kong et al., 2023); data deletion itself is susceptible to MIAs (Chen et al., 2021).

**Explaining the Disparate Susceptibility to Membership Inference Attacks:** Prior work has attributed the disparate impact of MIA mainly to the lack of generalization (or overfitting) (Yeom et al., 2018).

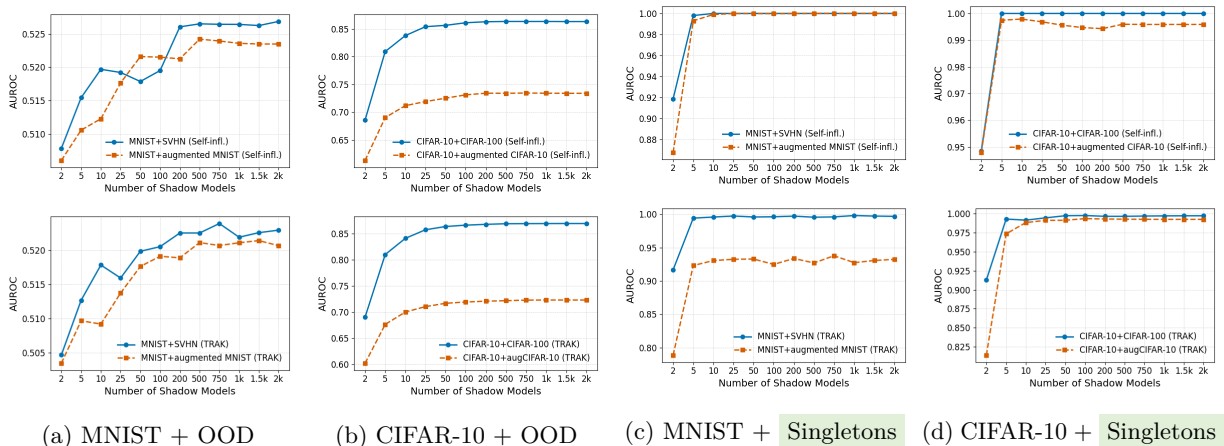

Figure 1: **AUROC of MIA by Carlini et al. (2022a) as a function of the number of shadow models. (a), (b):** Adversary picks challenge data randomly across over- and under-represented populations (*i.e.,* "All" in Table 1). **(c), (d):** Adversary picks challenge data from under-represented populations with high memorization scores (*i.e.,* "Under-represented"), identified via an *oracle*—an idealized assumption used here to measure the maximum potential benefit of memorization-guided sample selection. Memorization oracle instantiations: **(Top)** Empirical self-influence by Feldman & Zhang (2020); **(Bottom)** TRAK (Park et al., 2023).

However, these explanations fail to fully capture the nuanced nature of membership inference vulnerability. Overfitting, defined as the discrepancy between training and test performance, is often viewed as the root cause of MIA susceptibility. Yet, recent studies have shown that even models with good generalization can still memorize specific samples (Feldman, 2020). Memorization, in contrast to overfitting, refers to the extent to which an individual data point is uniquely stored by the model, making it particularly susceptible to MIAs. Our work highlights that memorization is a more fine-grained explanation for MIA vulnerability than overfitting, as memorized points remain identifiable even in well-generalized models. As empirical supporting evidence, in Appendix A, we show that most MIAs based on overfitting-based observations are not always effective; not *all samples* identified by these observations are highly susceptible to MIAs. Our theoretical framework refines this understanding by explicitly quantifying MIA success in terms of memorization.

### 5.2 Memorization

Memorization in machine learning is a well-studied phenomenon, particularly in the context of large models trained on diverse datasets (Feldman, 2020; Zhang et al., 2016). Traditional memorization scoring methods rely on influence functions (Koh & Liang, 2017), leave-one-out retraining (Feldman, 2020), and self-influence metrics (Feldman & Zhang, 2020). However, these methods are computationally expensive and impractical for large-scale models. To address this, recent works have developed computationally efficient alternatives. Gradient-based and trajectory-based methods, such as TracIn (Pruthi et al., 2020), track the influence of a sample over multiple epochs, reducing computational overhead while maintaining accuracy. Subsampling-based influence estimation techniques use Monte Carlo approximations of leave-one-out influence, allowing for efficient estimation without full retraining (Feldman, 2020). Linearization techniques, such as TRAK (Park et al., 2023), leverage first-order approximations to efficiently compute influence scores in large neural networks. Additionally, proxy-based approximations, including input loss curvurture (Ravikumar et al., 2024), have emerged as practical, scalable alternatives to exact memorization computation. These advancements allow for feasible per-sample memorization estimation in real-world settings, and can be directly used as successful instantiation of memorization oracle in our work.

**Connecting Membership Inference and Memorization:** Several prior works have explored connections between memorization and MIAs (Carlini et al., 2022b; Ye et al., 2022), particularly in identifying that some

data points are more susceptible to attacks. However, these works primarily remain empirical and lack a formal theoretical grounding that explicitly quantifies MIA success in terms of memorization. Our work advances this understanding by providing a rigorous theoretical bound on MI advantage as a function of memorization. Unlike prior works that rely on heuristic observations, we formalize the relationship through a hypothesis testing framework, proving that highly memorized samples provide a direct advantage to the adversary.

We highlight that, as memorization proxies become more efficient and accurate, our theoretical framework directly benefits from these advancements, enabling more effective and computationally feasible MIAs. By grounding our approach in theoretical guarantees while leveraging modern memorization estimation techniques, our work presents a principled and practical pathway for advancing membership inference research.

## 6 Conclusion

We attempt to explain the high efficacy of MIAs as a function of data's susceptibility to be memorized. Our main contribution is the first theoretical bound that connects label memorization and membership inference, using which we are also able to provide the first bound on the sample complexity needed for MIAs as a function of a sample's memorization score. More analysis is required to (a) convert the claims associated with mutual information from the work of Brown et al. (2021) to probability bounds (needed to instantiate attack strategies), (b) understand the efficacy of MI for samples that are unlikely to be memorized, and (c) instantiate more practical memorization oracles (i.e., those that are not computationally expensive). Further discussions on the implications of our work can be found in Appendix E.

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

## Appendix

An anonymized repository for our implementation is at `https://anonymous.4open.science/r/memorization-mi-24FF`.

## A   Motivational Experiments: Why Memorization?

Here, we aim to understand what properties of data explain the disparate performance of MIAs. This can be leveraged by the adversary to employ more successful attacks (as done in § 3 and § 4). Prior work shows that the disparate performance is associated with how atypical the sample is w.r.t other points in the training dataset. We wish to understand if this theory provides a nuanced (and holistic) understanding of MIA success with in-depth experiments.

**Experimental Setup:** We consider a training data $D^{\mathrm{tr}}$ consisting of two different datasets (say $D_{\mathrm{O}}^{\mathrm{tr}}$ and $D_{\mathrm{U}}^{\mathrm{tr}}$) with a mixing ratio $\alpha \in [0, 1]$ (*i.e.,* to obtain $D^{\mathrm{tr}} = \alpha \cdot D_{\mathrm{O}}^{\mathrm{tr}} + (1 - \alpha) \cdot D_{\mathrm{U}}^{\mathrm{tr}}$). In such settings, one dataset is an *over-represented group* (*i.e.,* $D_{\mathrm{O}}^{\mathrm{tr}}$ when the mixing coefficient is greater than 0.5), and the other is an *under-represented group* (*i.e.,* $D_{\mathrm{U}}^{\mathrm{tr}}$). Such composite datasets have been recognized in many contexts; for instance, in federated learning, data in many settings, data is pooled from different sources such as devices from various geographic regions Li et al. (2020); modern datasets are long-tailed and composed of varied subpopulations Zhu et al. (2014); Van Horn & Perona (2017); Babbar & Schölkopf (2019).

**Our notion of Atypicality:** We realize the under-represented group by picking data samples "far" from the over-represented group, which is commonly referred to as out-of-distribution (OOD) data. Commonly, this is realized by considering *covariate-shifted* OOD data (*i.e.,* data generated by applying semantics-preserving transformations to the training data, preserving the original labels), or *semantic* OOD data (*i.e.,* data having completely disjoint label sets from the normal training data) Bai et al. (2023).

In our experiment, the over-represented group is fixed as the entire dataset of MNIST LeCun et al. (2010) (60,000 labeled samples), while the choice of the under-represented group varies. We consider three variants: (**V1**) MNIST augmented by the approach proposed by Hendrycks *et al.* Hendrycks et al. (2022) (henceforth referred to as augMNIST), (**V2**) cropped SVHN Netzer et al. (2011), and (**V3**) CIFAR-10 Krizhevsky (2009). **V1** and **V2** correspond to covariate-shifted OOD, while **V3** is semantic OOD. To construct $D_{\mathrm{U}}^{\mathrm{tr}}$ (for a given variant), we choose a subset of samples (more details follow) from the corresponding training set; care is taken to ensure the consistent label assignment to both $D_{\mathrm{O}}^{\mathrm{tr}}$ and $D_{\mathrm{U}}^{\mathrm{tr}}$ [10]. All samples are (a) resized to match the dimension of MNIST samples ($28 \times 28 \times 1$), and (b) converted to grayscale[11]. We use the same CNN models as in the main experimental section (§ 4.1).

### A.1   Are All OOD Samples Susceptible?

In this subsection, we aim to examine if the OOD-ness of data is a sufficient explanation for susceptibility to MIAs. To check if OOD samples are more susceptible to MIAs, we choose a random subset from each of the aforementioned variants (**V1-3**) as our under-represented group ($D_{\mathrm{U}}^{\mathrm{tr}}$) and train models to achieve reasonable training and test accuracy (greater than 80% for train, and greater than 50% in most cases for test) on both groups. To do so, we need at least 1000 samples for augmented MNIST and cropped SVHN, and at least 6000 CIFAR-10 samples (10% of the total dataset); all these samples are chosen such that there are the equal number of samples for each class. Complete statistics (including the mixing ratio) are presented in Table 2.

In all the experiments, we first compute the privacy score which captures the vulnerability of a sample to an MIA (higher is more vulnerable) [12] We then compare the distributional pattern of privacy scores between over-represented and under-represented groups (blue and red histograms, respectively, in Figure 2a). We

---

[10]Consistent and balanced label assignment across classes; *i.e.,* all "airplane" images in CIFAR-10 are labeled as class 0 along with digit "0" images of MNIST etc.

[11]Gray Image = $0.2989 \times \mathrm{R} + 0.5870 \times \mathrm{G} + 0.1140 \times \mathrm{B}$.

[12]Privacy score $= \frac{|\mu_{\mathrm{in}} - \mu_{\mathrm{out}}|}{\sigma_{\mathrm{in}} + \sigma_{\mathrm{out}}}$ where $\mu$ and $\sigma$ are statistical parameters to characterize when a data sample is included (or excluded) in the training set. Refer to § VII-B of Carlini *et al.* Carlini et al. (2022a) for the full description.

| Over-represented | Under-represented | $\alpha$ | # Samples | Model | Train Acc. (%) | | Test Acc. (%) | |
|---|---|---|---|---|---|---|---|---|
| | | | | | $D_{\mathrm{O}}^{\mathrm{tr}}$ | $D_{\mathrm{U}}^{\mathrm{tr}}$ | $D_{\mathrm{O}}^{\mathrm{te}}$ | $D_{\mathrm{U}}^{\mathrm{te}}$ |
| | None | 1 | 0 | CNN32 | $98.54 \pm 0.08$ | N/A | $98.36 \pm 0.14$ | N/A |
| MNIST | augmented MNIST | 0.99 | 1000 | CNN32 | $99.15 \pm 0.13$ | $89.10 \pm 1.7$ | $98.43 \pm 0.19$ | $65.88 \pm 1.39$ |
| | SVHN | 0.99 | 1000 | CNN32 | $99.44 \pm 0.05$ | $90.07 \pm 0.66$ | $98.60 \pm 0.09$ | $64.76 \pm 1.31$ |
| | CIFAR-10 | 0.93 | 6000 | CNN64 | $99.27 \pm 0.86$ | $82.69 \pm 6.84$ | $98.18 \pm 0.47$ | $40.72 \pm 1.17$ |
| | None | 1 | 0 | ResNet-9 | $97.06 \pm 0.77$ | N/A | $90.02 \pm 0.02$ | N/A |
| CIFAR-10 | augmented CIFAR-10 | 0.98 | 1000 | ResNet-18 | $96.39 \pm 0.36$ | $86.85 \pm 2.14$ | $87.79 \pm 1.16$ | $54.41 \pm 1.79$ |
| | CIFAR-100 | 0.98 | 1000 | ResNet-9 | $96.59 \pm 0.21$ | $79.12 \pm 21.7$ | $88.39 \pm 0.59$ | $35.37 \pm 2.55$ |

Table 2: **Salient features of the datasets used in our experiments in § A.1.** Distance represents the distance between $D_{\mathrm{O}}^{\mathrm{tr}}$ and $D_{\mathrm{U}}^{\mathrm{tr}}$ (larger is more). Accuracy is reported in 95% confidence interval – for each iteration in bootstrapping, we include 70% of $D^{\mathrm{tr}}$ as the training set, and repeat 2000 times (as done by Feldman Feldman & Zhang (2020)). We also ensure that each sample in $D^{\mathrm{tr}}$ appears in the training set for half of the 2000 iterations, following the setup of Carlini *et al.* Carlini et al. (2022a).

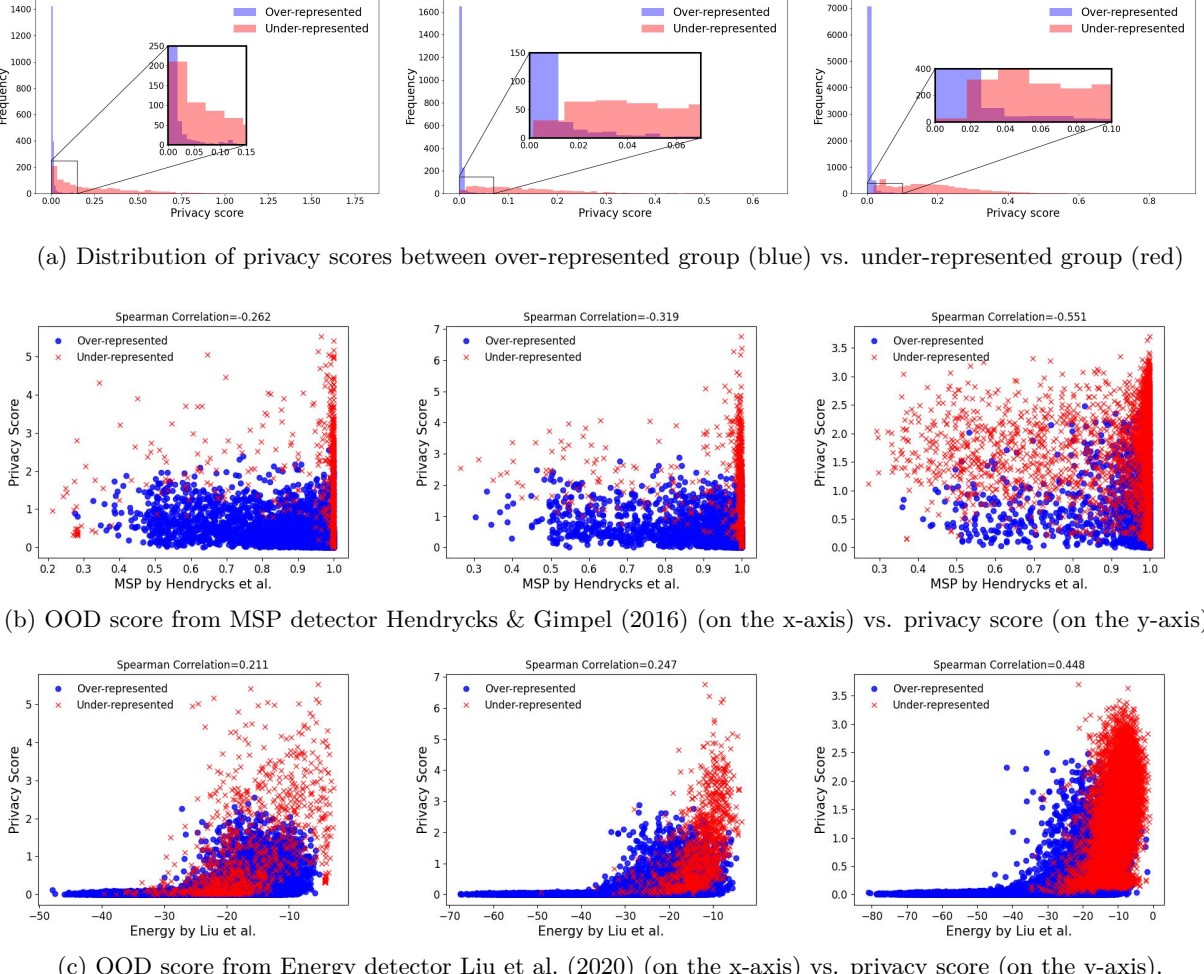

(a) Distribution of privacy scores between over-represented group (blue) vs. under-represented group (red)

(b) OOD score from MSP detector Hendrycks & Gimpel (2016) (on the x-axis) vs. privacy score (on the y-axis).

(c) OOD score from Energy detector Liu et al. (2020) (on the x-axis) vs. privacy score (on the y-axis).

Figure 2: **OOD data is not always more susceptible to MI attack than ID data.** (i) **L**: CNN32 trained on MNIST+augMNIST (**V1**), (ii) **M**: CNN32 trained on MNIST+SVHN (**V2**), (iii) **R**: CNN64 trained on MNIST+CIFAR-10 (**V3**).

also measure the correlation between the privacy score and OOD-ness of data, which is quantified via the output score from OOD detectors. Here we consider the two representative OOD detectors; MSP detector by

Hendrycks *et al.* Hendrycks & Gimpel (2016) (see Figure 2b) and Energy detector by Liu *et al.* Liu et al. (2020) (see Figure 2c). Higher value implies more OOD-ness.

**Observations:** Several important observations can be made. First, we want to understand if *all* OOD samples are highly susceptible to our MIA. *This is not the case.* In Figure 2a, while the majority of under-represented data (red histogram) has a higher privacy score than over-represented data (blue histogram), they are not completely separated from each other. Accordingly, the red points (representing under-represented data) and blue points (representing over-represented data) are overlapped in Figure 2b, and a large fraction of the under-represented samples are not highly susceptible to MIAs (*i.e.,* many OOD samples have the privacy score below 2 on the y-axis). Next, we want to examine the degree of correlation between being OOD and MIA susceptibility. We capture correlations using the Spearman's rank correlation coefficient with *p*-value less than 0.001 for statistical significance; (absolute) values less than 0.4 are considered to represent a weak correlation, values between 0.4 and 0.59 are considered to represent a moderate correlation, and values greater than 0.6 are considered to represent a strong correlation; the sign dictates whether the correlation is positive or not (irrelevant for our discussion). We observe that the absolute degree of correlation below 0.4 or at most 0.551 (see the numbers on the subfigure titles in Figure 2b, meaning that there is *no strong correlation* between OOD scores (measured by the approach of Hendrycks *et al.* Hendrycks & Gimpel (2016)) and MIA success; experiments with other OOD detector Liu et al. (2020) show similar trends (see Figure 2c).

**Implications:** The aforementioned observations strongly indicate that relying solely on OOD scores to detect membership status is not a definitive approach [13]. Among all OOD data points, only a subset of them exhibits high privacy scores (*i.e.,* more susceptibility to MI attacks). This suggests that while previous explanations on the disparate impact of MI vulnerability hold true to some extent, they are rather coarse and require further refinement. We posit that this is because the conventional definitions of OOD-ness found in the literature (*e.g.,* label disjointedness) are not precisely aligned with the correct characteristics of samples that semantically differentiate them from the majority of the data, thereby failing to explain their vulnerability to MIAs. In fact, current definitions of OOD-ness often suffer from flaws Bitterwolf et al. (2023), and the results obtained based on such assumptions lack reliability and precision.

### A.2 Are Memorized Samples Susceptible?

A second phenomenon that can cause disparate performance of MIAs is the model's capability of memorizing a sample (irrespective of it being from within or outside the training data distribution). Brown *et al.* Brown et al. (2021) note that the mutual information between a model's parameters and samples that are memorized is high; this leads us to believe that such samples are highly likely to be susceptible to MIAs. We proceed to identify the samples with high label memorization scores ($> 0.8$), calculated based on the definition in Eqn. 1 [14]. Note that by doing so, we reduce the size of the under-represented group (details in Table 3). We then measure the Spearman's rank correlation coefficient between the memorization score and the privacy score.

| Over-represented | Under-represented | # Samples | | Model | Train Acc. (%) | | Test Acc. (%) | |
| | | $|D_{\mathrm{O}}^{\mathrm{tr}}|$ | $|D_{\mathrm{U}}^{\mathrm{tr}}|$ | | $D_{\mathrm{O}}^{\mathrm{tr}}$ | $D_{\mathrm{U}}^{\mathrm{tr}}$ | $D_{\mathrm{O}}^{\mathrm{te}}$ | $D_{\mathrm{U}}^{\mathrm{te}}$ |
|---|---|---|---|---|---|---|---|---|
| | augmented MNIST | 60,000 | 219 | CNN32 | $99.27 \pm 0.14$ | $72.08 \pm 5.93$ | $98.44 \pm 0.20$ | $49.38 \pm 3.04$ |
| MNIST | cropped SVHN | 60,000 | 209 | CNN32 | $99.32 \pm 0.10$ | $71.40 \pm 6.22$ | $98.48 \pm 0.20$ | $19.67 \pm 2.82$ |
| | CIFAR-10 | 60,000 | 2153 | CNN64 | $99.43 \pm 0.35$ | $72.66 \pm 1.86$ | $98.28 \pm 0.31$ | $20.78 \pm 1.215$ |
| CIFAR-10 | augmented CIFAR-10 | 50,000 | 274 | ResNet-18 | $96.35 \pm 0.84$ | $71.02 \pm 6.21$ | $87.83 \pm 1.57$ | $50.13 \pm 2.02$ |
| | CIFAR-100 | 50,000 | 254 | ResNet-9 | $96.61 \pm 0.2$ | $69.69 \pm 6.1$ | $87.35 \pm 0.56$ | $11.21 \pm 1.6$ |

Table 3: **Salient features of the datasets used in our experiments in § A.2.** Accuracy reported in 95% confidence interval over 2000 iterations as in Table 2.

**Observations:** Several observations can be made from Figure 4a. First, the samples that have the high memorization scores (*i.e.,* the red points) roughly correspond to those samples which have a high privacy score in Figure 2b; this suggests that the OOD samples with high memorization scores are indeed those

---

[13] We utilize popular baselines from literature to capture this effect. We acknowledge that there are no perfect measures to determine OOD-ness, the OOD detector outputs are mere proxies of them.

[14] For the complete description of estimating privacy and memorization scores, see Algorithm 1 in Appendix C.2.

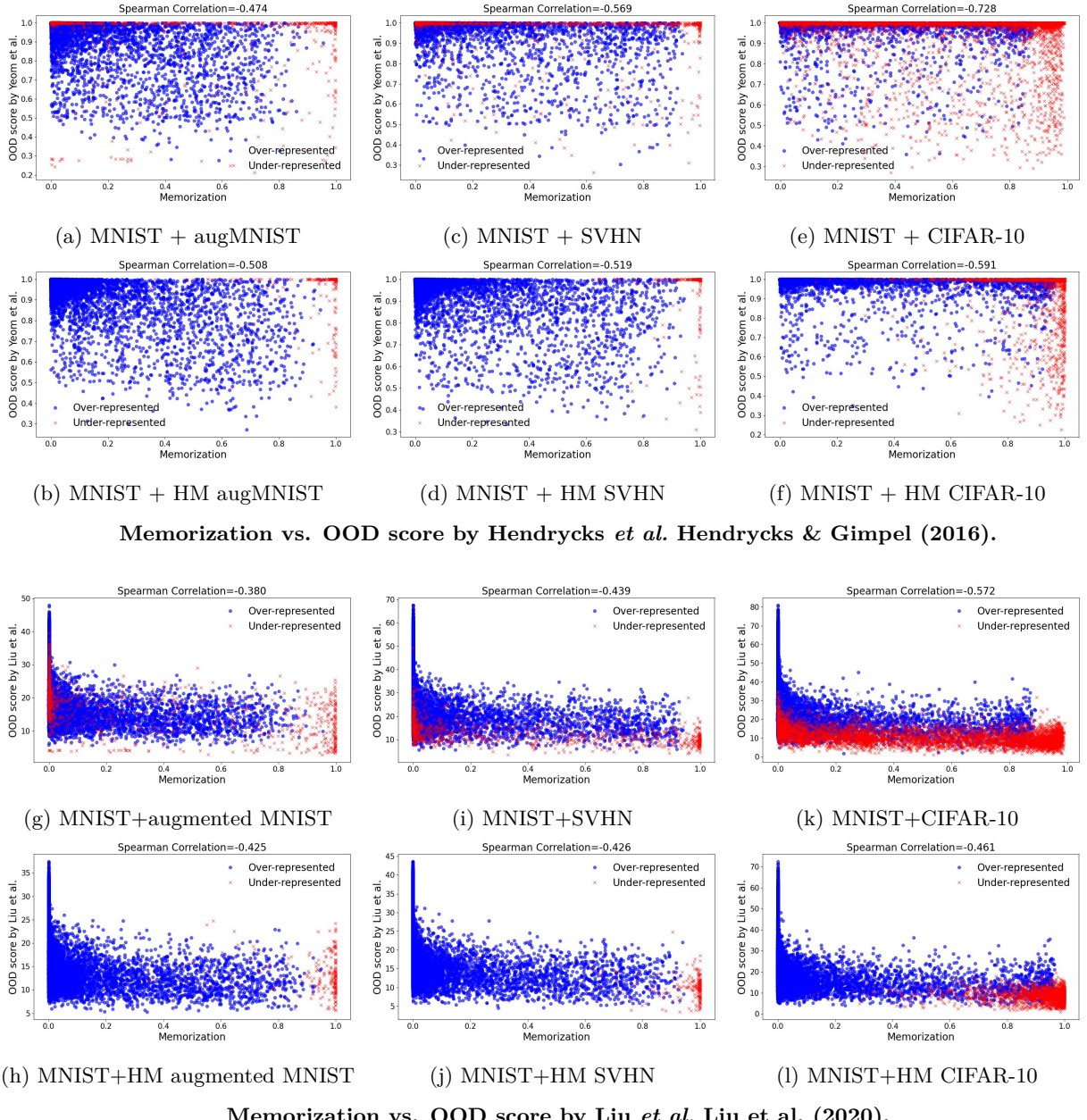

Figure 3: **Correlation between how likely to be memorized vs. being OOD.** An MI adversary constructs the dataset by varying the choice of samples for the under-represented population (random OOD in the first row vs. OOD with high memorization scores in the second row). "HM" refers to Highly Memorized. Memorization score is approximated by the empirical self-influence method in Feldman & Zhang (2020). We compute the OOD score with the two most popular baselines in the literature: Hendrycks *et al.* Hendrycks & Gimpel (2016) and Liu *et al.* Liu et al. (2020). For both of the choices, we observe a clearer separation between the populations (blue vs. red) in the second row than in the first row.

that are highly susceptible to MIAs. Next, we visualize the samples corresponding to different memorization scores from each dataset (see Figure 4b). It confirms our intuition: the samples from the over-represented group fall into the low-memorization regime, atypical samples from the over-represented group (and a few similar samples from the under-represented group) compose the mid-level memorization regime, and the

under-represented samples belong to the high-memorization regime. Finally, we observe that the correlation in this case is better; this suggests that memorization is a better indicator of MIA susceptibility.

**Implications:** The aforementioned observations suggests that memorization values are a stronger indicator of MIA susceptibility. This suggests that propensity to be memorized is a better characteristic of the data that can explain its susceptibility to MIAs.

> Prior work Carlini et al. (2022b) notes that outlier samples are more susceptible to MIAs. Through our analysis, we note that *not all outliers* are susceptible, and the current definition for "outliers" are unreliable. However, we observe that those outliers that are also highly likely to be memorized are definitively susceptible to MIAs. Thus, memorization provides a more nuanced understanding of the disparate MIA performance, and accurately characterizes MIA vulnerability.

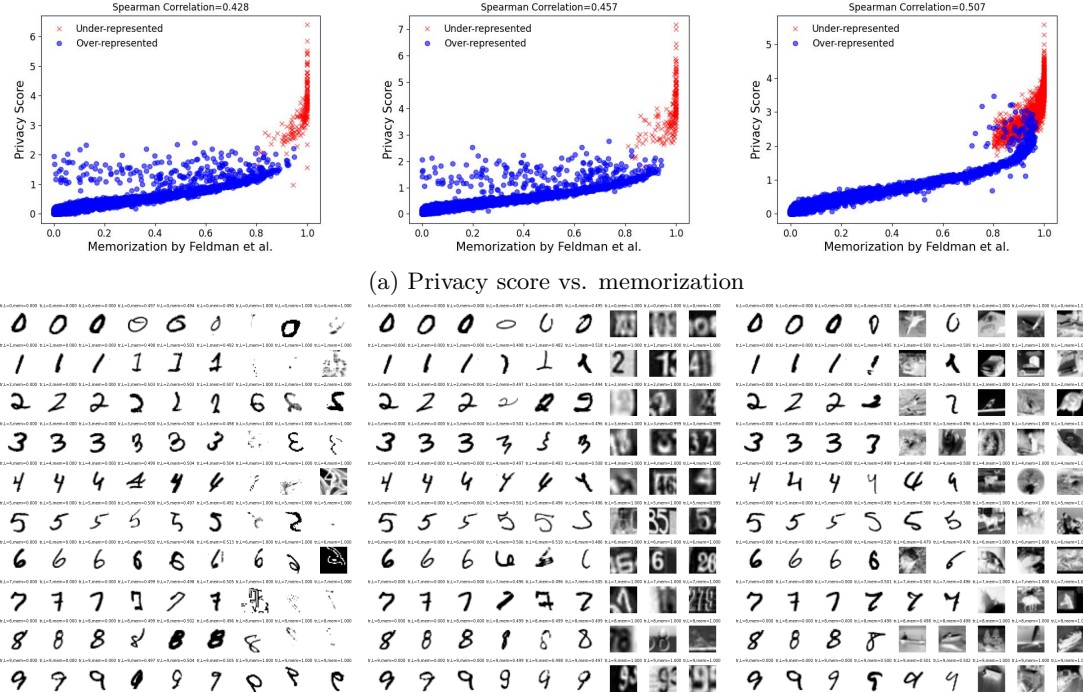

(a) Privacy score vs. memorization

(b) Random examples whose memorization value is close to 0 (first three columns), 0.5 (next three columns), 1 (last three columns). Each row shows images belonging to each of 10 classes.

Figure 4: (i) **L**: MNIST+augmented MNIST (**V1**), (ii) **M**: MNIST+SVHN (**V2**), (iii) **R**: MNIST+CIFAR-10 (**V3**)

## B   Theory

### B.1   Proofs of Equation 7

Our memorization definition in Equation 1 is a point-wise difference in marginal prediction behavior for the fixed input-label pair. In contrast, the Hellinger distance $H^2(P_0, P_1)$ is a global divergence between distributions $P_0$ and $P_1$ over the entire model space. It quantifies how distinguishable these two distributions are in totality. To relate the two, we consider the total variation distance:

$$TV(P_0, P_1) = \sup_A |\Pr_{P_0}[A] - \Pr_{P_1}[A]|$$

By choosing the measurable set $A = \theta : \theta(x) = y$, we obtain

$$TV(P_0, P_1) \geq \texttt{mem}(L, S, z)$$

since this choice of $A$ captures exactly the quantity defined as memorization. Now we introduce an additional assumption as follows.

**Prediction-Aligned Divergence Assumption**: The dominant change between the model distributions $P_0$ and $P_1$ arises from the shift in the model's predicted label on input z (*i.e.,* formally, the considered measurable set $A = \theta : \theta(x) = y$ is the primary contributor to the mass difference, such that

$$TV(P_0, P_1) = \texttt{mem}(L, S, z) \tag{8}$$

The above assumption implies that adding the training sample $z = (x, y)$ only affects the model's output for that specific prediction ($\theta(x) = y$) and has no other effects that would create a larger TV distance. We acknowledge that this is an idealized condition, but it allows for a tractable analysis that isolates the very mechanism many MIAs are designed to exploit. We believe this is a reasonable lens for theoretical examination, and our empirical findings, which demonstrate a strong link between high memorization scores and MIA success, support the practical relevance of focusing on this specific effect.

By plugging in the Equation 8 to the well-known relation between the TV distance and Hellinger distance (§ 2.3), we obtain $H^2(P_0, P_1) \leq \texttt{mem}(L, S, z)$.

### B.2   Theoretical Extensions

We have the following lemma, which will come in handy in several scenarios.

**Lemma 3** *Let $g$ be any bounded function from $\Theta \times \mathcal{X}$ to $\mathbb{R}$. Then there exists an adversary whose advantage is equal to:*

$$\Pr_{\theta \in L(S \cup \{z\})} g(\theta, z) - \Pr_{\theta \in L(S)} g(\theta, z)$$

*Let us call the above expression $\delta(g, L, S, z)$. We immediately have.*

$$Adv(L, \mathcal{A}) \quad \geq \quad \delta(g, L, S, z)$$

The proof structure is as follows: construct a probability distribution $p$ over $\Theta \times \mathcal{X}$ that is proportional to $f$ (*i.e.,* $p(\theta, z) = \alpha f(\theta, z).)$). Consider an adversary that outputs 1 with probability $p(\theta, z)$. Then computing the advantage is immediate.

Similarly, the following items also hold:

**C1.** Consider a bounded loss function $\ell$ with domain $\Theta \times \mathcal{X}$. Then we get the advantage as follows,

$$\Pr_{\theta \in L(S \cup \{z\})} \ell(\theta, z) - \Pr_{\theta \in L(S)} \ell(\theta, z)$$

Note that the expression given corresponds to the stability of the loss function on a sample $z$ with respect to a dataset $S$.

**C2.** Consider a bounded OOD detection function $ood$ with domain $\Theta \times \mathcal{X}$. Then we get the advantage as follows,

$$\Pr_{\theta \in L(S \cup \{z\})} ood(\theta, z) - \Pr_{\theta \in L(S)} ood(\theta, z)$$

Note that the expression given corresponds to the stability of the OOD detection function on $z$ with respect to a dataset $S$.

## C   Experimental Details

We run all experiments with Tensorflow, Keras, Jax and NVDIA GeForce RTX 2080Ti GPUs. To accelerate the shadow model training when necessary, we use PyTorch with FFCV dataloader Leclerc et al. (2022).

### C.1   Setup

**Datasets:** Here we describe the datasets used throughout our paper.

1. **MNIST** LeCun et al. (2010). The MNIST dataset consists of 60,000 handwritten digits (between 0 to 9) for train set, and 10,000 images for test set.

2. **SVHN** Netzer et al. (2011). SVHN is the color images of real world house numbers. We use the cropped version of the SVHN dataset, where images are tightly cropped around each digit of 0-9. From the original train/test set, we randomly select 1,000 images that are equally balanced across 10 labels.

3. **CIFAR-10** Krizhevsky (2009). We use randomly sampled 6,000 images from the original 50,000 CIFAR-10 training images. We preserve their original labels so that images for class $i$ of CIFAR-10 where $i \in [0, 9]$ are assigned to the same class as the MNIST images with label $i$.

4. **CIFAR-100** Krizhevsky (2009). We use randomly sampled 1,000 images from the original train set of 50,000 CIFAR-100 images. We use the first 10 classes out of 100 classes in total including 100 images each.

5. **augMNIST and augCIFAR-10**. We generate the augmented variation of the MNIST dataset by using the technique in Hendrycks et al. (2022). Original MNIST image is mixed with augmented versions of itself and the grayscale version of provided 14248 fractal images. We use severity level of 2 for mixing, and severity level $= 4$ for base augmentation operations (such as normalization, random cropping, and rotation) for 4 iterations for each image. We use 1,000 images randomly chosen from the augmented train set, while ensuring the balance across 10 classes.

**Models:** We adopt CNN architectures from Carlini *et al.* Carlini et al. (2022a): CNN models with 32 and 64 convolutional filters (referred to as CNN32 or CNN64, respectively). We also include two ResNet architectures (ResNet-9 and Resnet-18) He et al. (2016) into our consideration. We use them when training shadow models or target models for MIAs.

**MIAs.** We briefly describe the details of different membership inference attack methods.

1. Yeom *et al.* Yeom et al. (2018). The attack is based on the observation that ML models are trained to minimize the loss of the training samples, and the loss values on them are more likely to be lower than the samples outside of the training set. They suggest applying threshold on the loss values from the ML model to infer the membership of an input.

2. Shokri *et al.* Shokri et al. (2017). The attack uses a trained ML model to ascertain membership/non-membership. In our experiments, a fully-connected neural network with one hidden layer of size 64 with ReLU (rectifier linear units) activation functions and a SoftMax layer is used to distinguish feature vectors obtained from shadow models trained with and without a data-point.

3. Song *et al.* Song & Mittal (2021). The attack uses shadow models to approximate the distributions of entropy values, instead of cross-entropy loss. Given a target model and sample, they conduct hypothesis test between the member and non-member distributions for each class.

4. Sablayrolles *et al.* Sablayrolles et al. (2019). Their attack also utilizes the loss value. The loss is scaled for better attack accuracy, using a per-sample hardness threshold which is identified in a non-parametric way from shadow model training.

5. Carlini *et al.* Carlini et al. (2022a). Refer Algorithm 1 in Appendix C.2. We used the implementation provided by the authors for the attack.

> **Disclaimer:** Our evaluation setup spans 7 datasets and 4 models, and 5 MIAs. The magnitude of this evaluation is comparable to Carlini *et al.* Carlini et al. (2022a). We stress that performing such rigorous evaluation is also computationally expensive, requiring the training of a cumulative total of 38000 shadow models (2000 shadow models for each dataset and each variation of under-represented data selection).

### C.2 Algorithm for MIA and Memorization

We describe the algorithm proposed by Carlini *et al.* Carlini et al. (2022a) for MI (reproduced from their work), and Feldman and Zhang Feldman & Zhang (2020) for label memorization in Algorithm 1. Observe that both the MI success and label memorization can be obtained from the same algorithm by saving some state.

**Common Subroutines:** Notice that the privacy score estimation relies on training numerous models, some with the point under consideration, and some without (refer the grey box in Algorithm 1). We refer to this as the leave-one-out (LOO) subroutine. Observe that the exact same LOO subroutine can be used to empirically estimate memorization (which is also used to measure algorithmic stability and influence). As noted in § E, the attack by Carlini *et al.* Carlini et al. (2022a) couples this with hypothesis testing, leading to high success rates for all samples.

---

**Algorithm 1** Algorithm for estimating memorization, privacy score, and the outcome of Carlini et al. (2022b) attack.

**Require:** model $\theta$, example $z = (x, y)$, data distribution $\mathcal{D}$
1: $\text{confs}_{\text{in}} = \{\}$
2: $\text{confs}_{\text{out}} = \{\}$
3: $\mathcal{F}_{\text{in}} = \{\}$
4: $\mathcal{F}_{\text{out}} = \{\}$

5: **for** $m$ times **do**
6: $\quad \hat{S} \xleftarrow{\$} \mathcal{D}$ $\qquad\qquad\qquad\qquad\qquad\qquad\qquad\qquad\quad$ ▷ *Sample a shadow dataset*
7: $\quad \theta_{\text{in}} \leftarrow L(\hat{S} \cup \{(x, y)\})$ $\qquad\qquad\qquad\qquad\qquad\qquad\quad$ ▷ *train IN model*
8: $\quad \mathcal{F}_{\text{in}} \leftarrow \mathcal{F}_{\text{in}} \cup \{\theta_{\text{in}}\}$
9: $\quad \theta_{\text{out}} \leftarrow L(\hat{S} \backslash \{(x, y)\})$ $\qquad\qquad\qquad\qquad\qquad\qquad\;$ ▷ *train OUT model*
10: $\quad \mathcal{F}_{\text{out}} \leftarrow \mathcal{F}_{\text{out}} \cup \{\theta_{\text{out}}\}$
11: $\quad \text{confs}_{\text{in}} \leftarrow \text{confs}_{\text{in}} \cup \{\phi(\theta_{\text{in}}(x)_y)\}$
12: $\quad \text{confs}_{\text{out}} \leftarrow \text{confs}_{\text{out}} \cup \{\phi(\theta_{\text{out}}(x)_y)\}$
13: **end for**

14: $\mu_{\text{in}} \leftarrow \texttt{mean}(\text{confs}_{\text{in}})$
15: $\mu_{\text{out}} \leftarrow \texttt{mean}(\text{confs}_{\text{out}})$
16: $\sigma^2_{\text{in}} \leftarrow \texttt{var}(\text{confs}_{\text{in}})$
17: $\sigma^2_{\text{out}} \leftarrow \texttt{var}(\text{confs}_{\text{out}})$
18: $\text{confs}_{\text{obv}} \leftarrow \theta(x)_y$

19: $\Lambda = \frac{p(\text{confs}_{\text{obv}} | \mathcal{N}(\mu_{\text{in}}, \sigma_{\text{in}}))}{p(\text{confs}_{\text{obv}} | \mathcal{N}(\mu_{\text{out}}, \sigma_{\text{out}}))}$
20: $\widetilde{\text{mem}}(\mathcal{A}, S, z) := \text{Pr}_{\theta_{\text{in}} \in \mathcal{F}_{\text{in}}}[\theta_{\text{in}}(x) = y] - \text{Pr}_{\theta_{\text{out}} \in \mathcal{F}_{\text{out}}}[\theta_{\text{out}}(x) = y]$

21: **return** $\frac{|\mu_{\text{in}} - \mu_{\text{out}}|}{\sigma_{\text{in}} + \sigma_{\text{out}}}$ and $\widetilde{\text{mem}}(\mathcal{A}, S, z)$ and $\Lambda$

---

# D  Additional Results

## D.1  Attacks

| Method | Dataset ($D^{\text{tr}}$) | AUROC ↑ | | TPR @ 0.1% FPR ↑ | |
|---|---|---|---|---|---|
| | | All | Under-represented | All | Under-represented |
| Yeom *et al.* Yeom et al. (2018) | MNIST+CIFAR-10 (random) | 0.51 | 0.73 | 0.0 % | 0.0 % |
| | MNIST+CIFAR-10 (subgroup) | 0.50 | 0.80 | 0.0 % | 0.0 % |
| | MNIST+CIFAR-10 (singletons) | 0.51 | **0.96** | 0.0 % | 0.0 % |
| Shokri *et al.* Shokri et al. (2017) | MNIST+CIFAR-10 (random) | 0.56 | 0.85 | 0.66 % | 1.09 % |
| | MNIST+CIFAR-10 (subgroup) | 0.52 | 0.86 | 0.16 % | 0.98 % |
| | MNIST+CIFAR-10 (singletons) | 0.55 | **0.99** | 0.95 % | **16.52 %** |
| Sablayarolles *et al.* Sablayrolles et al. (2019) | MNIST+CIFAR-10 (random) | 0.54 | 0.87 | 0.20 % | 1.98 % |
| | MNIST+CIFAR-10 (subgroup) | 0.53 | 0.92 | 1.22 % | 14.95 % |
| | MNIST+CIFAR-10 (singletons) | 0.53 | **1.0** | 1.98 % | **21.50 %** |
| Song *et al.* Song & Mittal (2021) | MNIST+CIFAR-10 (random) | 0.54 | 0.87 | 0.20 % | 2.06 % |
| | MNIST+CIFAR-10 (subgroup) | 0.53 | 0.95 | 0.20 % | 6.09 % |
| | MNIST+CIFAR-10 (singletons) | 0.53 | **0.99** | 1.98 % | **15.53 %** |
| Carlini *et al.* Carlini et al. (2022a) | MNIST+CIFAR-10 (random) | 0.59 | 0.96 | 5.7 % | 44.15 % |
| | MNIST+CIFAR-10 (subgroup) | 0.52 | 0.97 | 0.27 % | 32.44 % |
| | MNIST+CIFAR-10 (singletons) | 0.56 | **1.0** | 4.58 % | **96.51 %** |

Table 4: **Additional comparison of representative MIAs.** For each choice of MIA and mixture dataset, we direct readers to compare AUROC and TPR between (i) (All) vs. (Under-represented), and (ii) (random) vs. (subgroup) vs. (singletons) to see the effect of utilizing singletons in MIAs. Here singletons are identified by the empirical self-influence algorithm in Feldman & Zhang (2020). Best results are **boldfaced**.

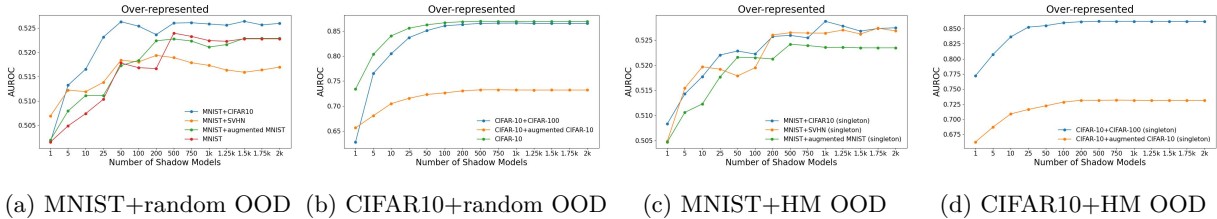

(a) MNIST+random OOD    (b) CIFAR10+random OOD    (c) MNIST+HM OOD    (d) CIFAR10+HM OOD

Figure 5: **AUROC of MIA by Carlini et al. (2022a) varying the number of shadow models.** Adversary picks challenge data from over-represented subpopulation. HM stands for "Highly Memorized."

When the adversary selects challenge points from the over-represented subpopulation—which dominates the training data when samples are chosen randomly—it needs to train over 1,000 shadow models for MNIST (Figure 5a and Figure 5c) and over 100 shadow models for CIFAR-10 (Figure 5b and Figure 5d). This requirement is significantly higher than what we observed in Figure 1, where the adversary chooses challenge points from an under-represented subpopulation. The reduction in the number of required shadow models can be even more substantial if the adversary selects data with high memorization scores. We also present the true positive rate (TPR) at 0.1% false positive rate (FPR) in Figure 6.

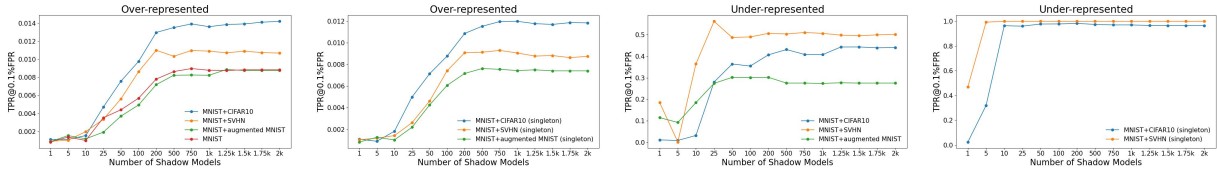

(a) MNIST+ramdom OOD, attack on random data    (b) MNIST+HM OOD, attack on random data    (c) MNIST+random OOD, attack on HM OOD data    (d) MNIST+HM OOD, attack on HM OOD data

Figure 6: **TPR at FPR= 0.1% of MIA by Carlini et al. (2022a) varying the number of shadow models.** Challenge data is either from over-represented subpopulations, or HM samples. HM stands for "Highly Memorized."

## D.2 Instantiating Memorization Oracle with TRAK Park et al. (2023) instead of empirical self-influence Feldman & Zhang (2020)

As can be found in Figure 7, the separation between over-represented vs. under-represented subpopulations is much clearer with respect to the memorization score (Feldman & Zhang, 2020) than w.r.t TRAK score (Park et al., 2023). This empirically implies that empirical self-influence score computed with the subsampling alogirhtm in Feldman & Zhang (2020) is a more accurate proxy for singleton identification. However, TRAK is an attribution approximation method with significantly reduced computation overhead, and in this section, we show that one could still launch effective and efficient membership inference with TRAK as a memorization oracle. It may not be as good as the subsampling approach by Feldman & Zhang (2020), still substantially better than when launching MIAs with random (OOD) samples.

## E    Discussion

**Connections between LiRA and Label Memorization:** The work of Carlini *et al.* Carlini et al. (2022a) (LiRA) demonstrates state-of-the-art performance for MIAs, especially at low FPR regimes. To understand why, we urge the reader to analyze Algorithm 1 in their work. This approach is very similar to estimating memorization, as defined in Equation 1. The notable difference is line 4, where at each iteration, a dataset is sampled from the data distribution $\mathcal{D}$ (which does not happen for estimating memorization). However, upon analyzing their code [15], we observed that the implementation is *not faithful* to the algorithm. In particular,

---
[15]https://github.com/tensorflow/privacy/blob/master/research/mi_lira_2021/train.py#L215:L228

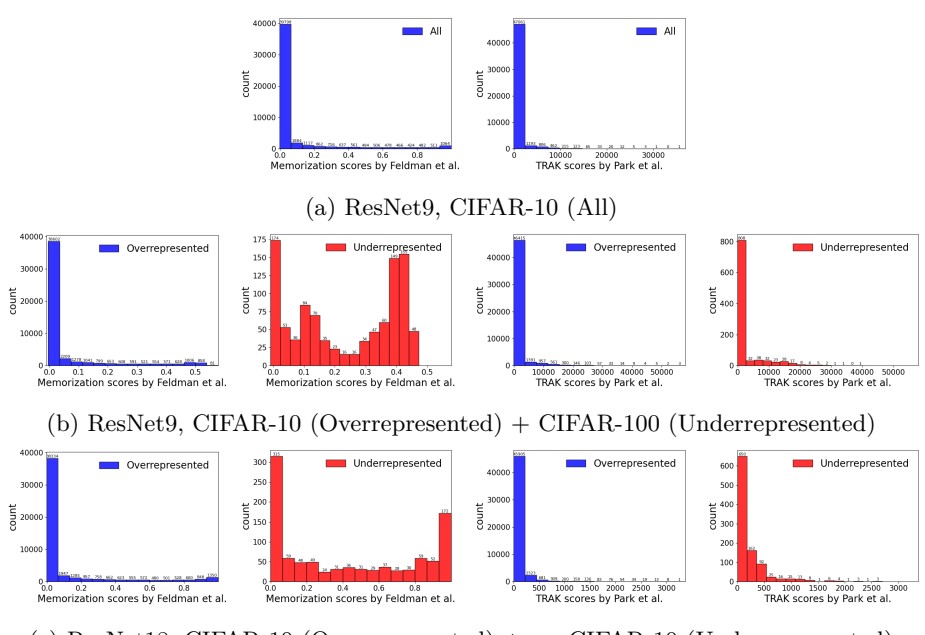

(a) ResNet9, CIFAR-10 (All)

(b) ResNet9, CIFAR-10 (Overrepresented) + CIFAR-100 (Underrepresented)

(c) ResNet18, CIFAR-10 (Overrepresented) + augCIFAR-10 (Underrepresented)

Figure 7: **Comparison of score distributions between Memorization (Feldman & Zhang, 2020) vs. TRAK (Park et al., 2023). first two figures in each row will be replaced with memorization distribution figures**

the implementation draws random subsets of the *training data* (not the data distribution), and thus the LiRA attack *is equivalent to estimating* label memorization (with some additional hypothesis testing). The high success of their LiRA approach is yet another testament to our connection with memorization.

**Connections between memorization and Differential Privacy (DP):** Our definition of memorization in Equation 1–which quantifies how much the distribution over learned models changes when a sample is added to the training set—-shares important conceptual parallels with DP. Here we discuss the connection in details.

- *Connection to Per-Sample Privacy Leakage:* The memorization score we analyze, introduced by Feldman & Zhang (2020), measures the change in behavior of a learning algorithm when a single sample $z$ is added to or removed from the training dataset $S$. This score can be interpreted as a *per-sample influence quantity.* In the DP literature, related notions of per-example privacy leakage aim to capture the sensitivity of an algorithm's output distribution to the inclusion of a single sample. Examples include the *privacy loss random variable* (the pointwise log-likelihood ratio), *per-instance Rényi divergence*, and *maximal individual leakage* (Ghazi et al., 2021; Gopi et al., 2021; Feldman, 2020; Feldman & Zrnic, 2021). While the memorization score is not equivalent to traditional DP metrics, such as $(\epsilon, \delta)$-DP, it is closely related in spirit. Instead of bounding pointwise log-ratios, the memorization score corresponds to a hypothesis test distinguishing the output distribution of $L(S \cup \{z\})$ from that of $L(S)$, evaluated on a specific measurable set: the event that a model predicts label $y$ on input $x$. This perspective aligns with recent work in DP that seeks more interpretable definitions of per-sample leakage, grounded in actual adversarial advantage.

- *Distinction from DP-based generalization or risk bounds:* While DP focuses on worst-case robustness over all possible neighboring datasets, our memorization score is instance-specific, capturing how a particular $z$ affects the model distribution. This makes our framework more granular and empirically tractable. Moreover, our results focus not only on bounding adversary success (as is typical in DP), but also on enabling more efficient MIAs when high-memorization points are known. Theorem 2 and

Corollary 1 formalize this by linking memorization directly to sample complexity, a direction that is orthogonal to, but inspired by, classical DP risk bounds.

**DP as a Defense:** DP is noted to be a promising defense against MIAs. We wish to study if this is the case for our "memorization-aware" attack as well. We use the exact same data (*i.e.,* same 1000 SVHN images out of the entire SVHN training data) and exact same subsampling (*i.e.,* 70% out of the entire training set for every shadow model) as normal training cases in § 4. We set target $\delta = 10^{-5}$, train every shadow model for 100 epochs with noise multiplier $= 1.3$, $\ell_2$ clipping norm $= 1.0$, resulting $\varepsilon = 3.2$. Such a small privacy budget results in low utility models (which display poor generalization). From Figure 8, we can see that when the model is trained with DP-SGD, samples that belong to the under-represented population also have low memorization scores (*i.e.,* samples are not memorized). Feldman (2020) notes that DP hampers memorization, and we observe a similar effect.

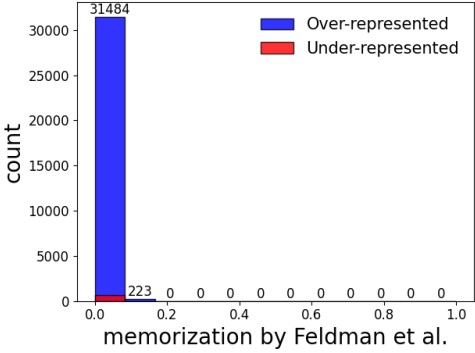

Figure 8: Memorization scores obtained when training using DP-SGD on a dataset comprised of MNIST + SVHN.

**Failed Inverse Generation Experiments:** Most of the MIAs attempt to ascertain distributional difference by training numerous shadow models, an expensive process. Both Brown *et al.* Brown et al. (2021) and Feldman Feldman (2020) note that (a) models memorize samples that belong to low density subsets of the data manifold, and (b) memorization implies strong test performance on other samples from the same low density subset. To this end, we wished to design an MIA for highly memorized samples based on the aforementioned phenomenon. To this end, we trained a flow-based model Dinh et al. (2016) on the CIFAR-10 dataset. Inverse-flow models (denoted by functions $g^{-1}$ and $g$ s.t. $x = g \circ g^{-1}(x)$ for an input $x$) are able to find the corresponding latent vector (on the data manifold) for an input; we believed that such a model would be able to accurately identify the low density subset that a given sample belongs to. The procedure was as follows: for a sample $z = (x, y)$ that has a high memorization score, find the corresponding latent $l = g^{-1}(x)$. Then, we perturb the latent vector (*i.e.,* find other vectors within an $\delta$ threshold) to obtain a modified latent vector $l'$, which can converted to a modified input using the flow model *i.e.,* $x' = g(l')$. Ideally, evaluating the test accuracy of the model on $x'$ would help understand if $z$ is memorized *i.e.,* if accuracy is high, then w.h.p $z$ is memorized. Despite ensuring a performant inverse-flow model, this approach failed because we are unable to ascertain the ground truth label for the modified input *i.e.,* if label of $x$ is $y$, we are unable to prove that the label of $x'$ is also $y$. While we believe this is a promising direction, and has been explored in the context of MIAs using generative adversarial networks Rezaei & Liu (2022), more experimentation is needed before inverse-flow models can be used in conjunction with memorization for creating an MIA.

**Can OOD-ness be a Memorization Proxy?** Our proposal requires the existence of a memorization oracle $\mathcal{O}_{mem}$. Practically, this can be instantiated by running the algorithm specified by Feldman and Zhang Feldman & Zhang (2020), which is a computationally expensive procedure. Recall that the results from § 4.3 suggest that computational overheads can be reduced even for OOD samples. One naturally wonders if the degree of being an outlier (measured by the OOD score) can serve as a proxy for identifying samples with high memorization scores *i.e.,* are samples with high outlier scores those which are highly likely to be memorized? We plot the correlation between OOD-ness and memorization score in Appendix A, and

observe that no such correlation exists. This suggests that the characteristics captured by OOD detectors in the status quo are not the same as those captured by label memorization. Definitions that are tied to stability are more likely to capture this effect (refer Appendix B.2). Thus designing inexpensive mechanisms to determine memorization scores remains an open problem.

**The Privacy Onion Effect:** The definition of label memorization is implicitly dependent on the dataset (see Equation 1). Carlini et al. (2022b) note that if a set of highly memorized points is removed from a dataset, points which previously had low memorization values now have high values (and vice versa). They term this the *privacy onion effect*. While defenders against memorization-guided MIAs may consider removing those samples that are likely to be memorized from the training dataset, two problems may emerge. The first is that the error of the model (on the low density subset associated with the deleted singleton) shoots up. Secondly, there will be a new set of points which are highly likely to be memorized.

