# OpenReview forum: "Exploring Connections Between Memorization And Membership Inference"
_TMLR — Rejected by TMLR_

### Review · Reviewer_MYbu · 2025-06-24

**Summary Of Contributions:**

This paper investigates the connection between sample memorization and privacy risk under membership inference (MI) attacks. By framing MI as a hypothesis testing problem, the authors derive theoretical results showing that the advantage of a strong adversary in performing an MI attack is equal to the sample’s memorization score. Consequently, samples with higher memorization scores are more vulnerable to MI attacks. In the more general setting—where the adversary does not have access to the true dataset—the required number of shadow models is shown to be bounded above and below by the inverse of the memorization score. Thus, fewer shadow models are needed to attack samples with higher memorization. The theoretical findings are validated empirically across multiple benchmarks and MI attack strategies.

**Audience:**

Yes

**Broader Impact Concerns:**

No, I don't think the paper posses any significant ethical implications and does not need a broader impact statement.

**Claims And Evidence:**

No

**Requested Changes:**

Please refer to the *Weaknesses* and *Questions* section above. While I believe all the concerns raised are important, I consider the issues related to the theoretical claims to be critical.

**Strengths And Weaknesses:**

**Strengths**

- The empirical validation of the theoretical claims is thorough. The authors evaluate an extensive range of MI attacks from the literature, demonstrating the robustness of their findings. However, the benchmark datasets could be further expanded, as discussed in the weaknesses section. Additionally, the experiments in Appendix A—comparing the predictive power of memorization scores versus OOD detection scores for privacy risk—are insightful and help motivate the problem formulation.

- The paper is generally well-written, with clear presentation of results through good figures. The authors also provide thoughtful explanations for observed trends, such as why the OOD subpopulation case yields stronger results than the singleton case for the CIFAR dataset. However, the presentation of the theoretical results could benefit from improvements, as outlined in the weaknesses section.

- The theoretical contributions appear novel, although I acknowledge that I am not deeply familiar with all aspects of the background literature in this area, so I may not be best positioned to fully assess the originality.


**Weaknesses**

- **I have a major concern regarding the correctness of Corollary 1**. It claims that the sample complexity (i.e., the total number of shadow models required) for the optimal adversary is both upper and lower bounded by $\frac{1}{mem(L,S,z)}$. Can the authors provide a clear derivation for this? Here is my attempt at it. Starting from equation (8), $H^2(P_0, P_1) \leq mem(L,S,z)$, we obtain $\frac{1}{H^2(P_0, P_1)} \geq \frac{1}{mem(L,S,z)}$. Then, by Lemma 2, if the sample complexity scales as $\Theta(\frac{1}{H^2(P_0, P_1)})$, his would only yield the following lower bound:  $\Omega(\frac{1}{mem(L,S,z)})$. This does not justify an upper bound of the form  $O(\frac{1}{mem(L,S,z)})$, which would be necessary to claim tightness and conclude that higher memorization leads to lower sample complexity.  I recommend the authors either provide a detailed derivation to justify this upper bound, or revise the corollary to more accurately reflect what is supported by the analysis.

- The theoretical contributions would benefit a lot from greater precision in their presentation. Theorem 2, for instance, is stated without specifying that it applies only to the strong adversary setting—i.e., one that has access to the full dataset. As currently written, the theorem appears to apply to any adversary, which is misleading. The statement should explicitly clarify the assumptions under which it holds. Similarly, Corollary 1 would be clearer if it explicitly restated the relevant notation and assumptions, rather than presenting the result as a single expression.  In addition, I recommend that the authors formally derive the inequality $H^2(P_0, P_1) \leq mem(L,S,z)$ from Feldman (2020), as it is central to the theoretical result. Including a proof or atleast a proof sketch in the appendix would also be helpful for readers who are not familiar with the original result, especially since this bound plays a crucial role in the derivation of the sample complexity claims.

- The authors could strengthen their empirical validation by incorporating subpopulation shift benchmarks such as those provided by SubPopBench (https://subpopbench.csail.mit.edu/), which offer predefined datasets with clearly defined overrepresented and underrepresented groups. This would eliminate the need to manually construct datasets with varying levels of group imbalance. These benchmarks are widely used in the community to evaluate model robustness under subpopulation shifts, and they directly assess a model's ability to learn from imbalanced group data. Leveraging them would make the empirical evaluation more standardized and further support the paper’s theoretical claims in a realistic and widely accepted setting.


**Questions**

- I don't follow the justification behind equation (6), how do we derive the lower bound?

- The notion of memorization as defined in Equation (1)—which captures the difference between a model trained with a particular sample
(*z*) and one trained without it—bears a strong resemblance to the core idea behind differential privacy (DP). In DP, the goal is to ensure that the output distribution of an algorithm does not change significantly when any single sample is added or removed, typically formalized via bounds on divergences like the KL divergence or Rényi divergence between neighboring datasets. Given this similarity, it would be valuable for the authors to clarify the connection between their notion of memorization and differential privacy. Specifically,

  * Does the proposed memorization score correspond to any known notion of per-sample privacy leakage or influence in the DP literature?
  * Are there existing results in the differential privacy community that relate per-sample divergence to membership inference risk or sample complexity?
  * How do the theoretical results in this paper (e.g., Theorem 2, Corollary 1) relate to or differ from known bounds in differential privacy, especially those concerning adversarial advantage or generalization?

---

> ### Author Response · Authors · 2025-07-24
> **Thanks for the detailed comments! Responses enclosed (part 1)**
>
> **Derivation of Corollary 1**:
>
> We appreciate the reviewer for pointing out a subtle but crucial issue. While our text noted that constants are absorbed in the $\tilde{\Theta}$ notation, we recognize that this implies a level of tightness that is not fully justified by our derivation. Specifically, as the reviewer notes, Equation (8) only implies a lower bound on the sample complexity in terms of $\texttt{mem}(L, S, z)$.
>
> Accordingly, we will revise Corollary 1 to state only the proven lower bound:
> $$
> \text{Sample complexity} = \Omega\left( \frac{1}{\texttt{mem}(L, S, z)} \right)
> $$
> We will also clarify in the surrounding discussion that while empirical results suggest this bound is tight in practice, a theoretical upper bound remains an open direction.
>
> Nonetheless, we note that this change is a reduction in the elegance of a tight characterization in Corollary 1, but the core message and the primary contribution are still preserved. The lower bound $\Omega\left( \frac{1}{\texttt{mem}(L, S, z)} \right)$ still formally and rigorously proves the fundamental relationship we set out to explore: a key ingredient ($\texttt{mem}(.)$) reduces the minimum cost of an attack represented in terms of sample complexity.
>
> **Clarity in presentation**
> We agree that several theoretical results would benefit from clearer presentation and will revise the manuscript as follows.
>
> - Theorem 2: We will revise the statement of Theorem 2 to explicitly note that it applies to a strong adversary who has access to the full training dataset $S$. This assumption is critical to the argument, and we appreciate the reviewer pointing out that it must be made explicit in the theorem body.
> - Corollary 1: We will revise Corollary 1 to restate the definitions of the distributions $P_0, P_1$, the meaning of $\texttt{mem}(L, S, z)$, and the setup for the hypothesis test. This will make the result more readable and reduce ambiguity about its scope and assumptions.
> - Derivation of $H^2(P_0, P_1) \leq \texttt{mem}(L, S, z)$: This inequality plays a key role in our theoretical results. While it follows from Feldman (2020), we agree that including a self-contained proof sketch would benefit readers unfamiliar with that work. We will add a derivation to the appendix that makes this step clear, including any assumptions about the model class or loss structure required for the inequality to hold.
>
> **Derivation of Equation 6**
> The inequality in Equation (6) arises from how the advantage of an adversary is defined in terms of their ability to distinguish between two training scenarios. Specifically, the advantage is defined as:
> $$
> \text{Adv}(L, \mathcal{A}) = \Pr(b_{\mathcal{A}} = 1 \mid b_{\mathcal{C}} = 1) - \Pr(b_{\mathcal{A}} = 1 \mid b_{\mathcal{C}} = 0)
> $$
> Here, $b_{\mathcal{C}}$ is a bit indicating whether the model was trained on $S \cup \{z\}$ or on $S$, and $b_{\mathcal{A}}$ is the adversary’s prediction of this bit.
>
> Now, suppose the adversary$\mathcal{A}$directly operates on the sampled model$L(S)$ or $L(S \cup {z})$ and returns 1 if it believes $z$ was included in training. Then:
>
> $$
> \Pr(b_{\mathcal{A}} = 1 \mid b_{\mathcal{C}} = 1) = \Pr(\mathcal{A}(L(S \cup \{z\})) = 1), \\
> \Pr(b_{\mathcal{A}} = 1 \mid b_{\mathcal{C}} = 0) = \Pr(\mathcal{A}(L(S)) = 1)
> $$
> Therefore, the advantage becomes exactly the difference in probabilities shown in Equation (6). Since the definition of advantage is a supremum over all adversaries, this particular construction provides a valid lower bound on $\text{Adv}(L, \mathcal{A})$. We will clarify this justification more explicitly in the revised version of the paper to avoid confusion.

---

> > ### Author Response · Authors · 2025-07-24
> > **Part 2**
> >
> > **Connections to DP**
> > We appreciate the reviewer drawing attention to the connection between our notion of memorization and the guarantees offered by differential privacy (DP). Indeed, our definition of memorization in Equation (1)—which quantifies how much the distribution over learned models changes when a sample $z = (x,y)$ is added to the training set—shares important conceptual parallels with DP. We respond to the reviewer’s sub-questions below:
> >
> > - Connection to per-sample privacy leakage in DP: The memorization score we analyze—originally introduced by (Feldman, 2020)—measures the change in behavior of a learning algorithm when a single sample $z = (x, y)$ is added to or removed from the training dataset. Formally, it is defined as the absolute difference in the probability of predicting $y$ on input $x$ under the two training regimes: $L(S)$ vs. $L(S \cup \{z\})$. This score can be interpreted as a per-sample influence quantity. In the DP literature, related notions of per-example privacy leakage or privacy loss aim to capture the sensitivity of an algorithm’s output distribution to the inclusion of a single sample. Examples include the privacy loss random variable (pointwise log-likelihood ratio), per-instance Rényi divergence, and maximal individual leakage (Feldman (2020) ; Gopi et al. (2021) ; Ghazi et al. (2021)). While Feldman’s memorization score is not equivalent to traditional DP metrics, such as $(\varepsilon, \delta)$-DP or per-instance Rényi DP, it is closely related in spirit. Instead of bounding pointwise log-ratios, the memorization score corresponds to a hypothesis test distinguishing $L(S)$ from $L(S \cup \{z\})$, evaluated at a specific measurable set: the event that a model predicts $y$ on input $x$. This perspective aligns with recent work in DP that seeks more interpretable and operational definitions of per-sample leakage grounded in actual adversarial advantage or influence on predictions.
> >
> > - Existing results relating divergence to MIA risk: Several works in both the differential privacy and generalization literature formally relate *membership inference risk* to *statistical divergences* such as total variation, Kullback–Leibler, or Rényi divergence—between the output distributions of a learning algorithm when trained on neighboring datasets $S$ and $S \cup \{z\}$. These divergences serve as upper bounds on the adversary’s advantage in a hypothesis test distinguishing whether $z$ was included in the training set (Yeom et al. (2018), Feldman (2020) , Ghazi et al. (2021)). More concretely, Yeom et al. (2018) relate generalization bounds and MIA risk to per-sample DP stability (e.g., upper bounds on adversary advantage in terms of privacy parameters). While Feldman and Zrnic (2021) primarily focus on per-sample Rényi privacy guarantees, classical results from hypothesis testing theory (e.g., Tsybakov (2008)) show that the adversarial advantage in distinguishing $L(S)$ vs. $L(S \cup \{z\})$—as in MIA—is upper-bounded by total variation distance, and by extension, the squared Hellinger distance (in certain conditions, as shown in our work). These divergences thus provide general bounds on MIA efficacy, even when specific assumptions like logit-Gaussianity (as in LiRA) do not hold. Our work further builds on this connection and derives explicit sample complexity bounds and by quantifying how memorization score governs adversarial efficiency.
> >
> > - Distinction from DP-based generalization or risk bounds: While DP focuses on worst-case robustness over all possible neighboring datasets, our memorization score is instance-specific, capturing how a particular $z$ affects the model distribution. This makes our framework more granular and empirically tractable. Moreover, our results focus not only on bounding adversary success (as is typical in DP), but also on enabling more efficient MIAs when high-memorization points are known. Theorem 2 and Corollary 1 formalize this by linking memorization directly to sample complexity, a direction that is orthogonal to, but inspired by, classical DP risk bounds.
> >
> > We will clarify these connections and distinctions more clearly in the revision, including references to DP-adjacent work and a brief discussion in Section 5 (related work)

---

> > > ### Author Response · Authors · 2025-07-24
> > > **Part 3 (final)**
> > >
> > > **Other benchmarks**
> > > We appreciate the reviewer for bringing the SubPopBench dataset to our attention. We agree that this standardized benchmark is valuable. We will try to include experiments using it in our revised manuscript, analyzing memorization scores for scores for samples from its predefined over- and underrepresented subgroups.
> > >
> > > **References**:
> > >
> > > - Alexandre B. Tsybakov. 2008. Introduction to Nonparametric Estimation (1st. ed.). Springer Publishing Company, Incorporated.
> > > - Gopi, Sivakanth, Yin Tat Lee, and Lukas Wutschitz. "Numerical composition of differential privacy." Advances in Neural Information Processing Systems 34 (2021): 11631-11642.
> > > - Feldman, Vitaly, and Tijana Zrnic. "Individual privacy accounting via a renyi filter." Advances in Neural Information Processing Systems 34 (2021): 28080-28091.
> > > - Yeom, Samuel, et al. "Privacy risk in machine learning: Analyzing the connection to overfitting." 2018 IEEE 31st computer security foundations symposium (CSF). IEEE, 2018.
> > > - Ghazi, Badih, et al. "Deep learning with label differential privacy." Advances in neural information processing systems34 (2021): 27131-27145.
> > >
> > > *In conclusion: thanks again for the excellent feedback! We are happy to respond to any further questions/concerns."

---

> > > > ### Author Response · Authors · 2025-07-28
> > > > **Ping!**
> > > >
> > > > Hello,
> > > >
> > > > We are happy to answer any further questions you may have. We understand that this period conflicts with the NeurIPS discussion period, but we are happy to engage with any further comments.

---

> > > > > ### Author Response · Authors · 2025-08-01
> > > > > **Ping x 2!**
> > > > >
> > > > > Hello,
> > > > >
> > > > > As we are nearing the end of the discussion period, we are writing to inquire if our responses have answered your questions. We are happy to answer more questions you may have, if you give us sufficient time to respond.
> > > > >
> > > > > Thanks for your time!

---

> > > > > > ### Comment · Reviewer_MYbu · 2025-08-01
> > > > > >
> > > > > > Thanks for the detailed response, I don't have any further questions. Please update the draft based on the discussion above, especially regarding the theoretical results and the relationship with DP. I believe the paper's theoretical contributions are significantly affected due to the incorrect proof for the upper bound on sample complexity ( $O(\frac{1}{mem(L,S,z)}$)). I encourage the authors to spend more on this. Thanks a lot again for the rebuttal!

---

### Review · Reviewer_uCxk · 2025-07-19

**Summary Of Contributions:**

This work proposes a new analysis of membership inference attacks (MIA) through the lens of label memorization. The authors show that samples with high memorization scores are easier to attack. The authors then provide an extensive experimental evaluation on MIA success for samples which are easily memorizable.

**Audience:**

Yes

**Claims And Evidence:**

No

**Requested Changes:**

1. The authors should revisit the main research questions of the paper and ensure that both the theorem statements and experiments are indeed answering those questions.
2. The authors should revisit the current results and think about how they could fit into the literature. For example, consider which gaps in the field each theorem is filling.
3. The authors should either provide clear reasoning to connect the empirical and theoretical results or consider a new, improved set of experiments that support the main results from the previous section.

**Strengths And Weaknesses:**

## Strengths:

1. The authors attempt to tackle a core problem in the field: Understanding and analyzing factors that lead to increased vulnerability of certain training data samples to privacy attacks on machine learning.
2. Extensive experimentation comparing existing attacks under different settings

## Weaknesses:

1. Main thesis is tautological and main results seem to use circular reasoning
   - E.g. (p. 4-5) We can restate the main thesis as:  “If a sample z is *always* classified correctly when it's IN and *never* classified correctly when it's OUT (i.e. Pr[M(z) = y | z is IN] = 1 and Pr[M(z) = y | z is OUT] = 0), then an adversary can successfully determine its membership.”
    - (p. 7) RQ2 states: *“If the adversary knows that a particular sample is likely to have a high memorization score, can they launch more computationally efficient MIAs?”*
        - If an adversary knows a particular subset of the data with high memorization scores, or high distinguishability between P_0 and P_1, then they should already be able to mount a concrete attack. The most expensive part of any MIA is finding a good proxy to the memorization score in the first place.

2. The paper misunderstands concrete MIA on machine learning from the literature and the gap between theory of MIA and existing attacks.
    - (p. 5-6) e.g. The authors state that existing work (LiRA) makes a parametric assumption on the distributions P_0, P_1. However, this work makes a parametric assumption (Gaussianity)  on the *logit (or prediction) distributions* of models produced by world 0 or world 1, rather than on the distribution of models themselves. Furthermore, the detailing of the strategy used in Scenario 2 is imprecise.

3. The notation throughout is dense and imprecise; Theorems are underexplained and/or difficult to follow
    - E.g. (p. 6) There is no elaboration of the cryptographic argument used and the jump from derivation to Theorem 2 is confusing. The result ends up being a restatement of the central tautology: Adversaries designed to detect memorization work well if the data was memorized

4. Experiments lack a concise connection to the analysis, and the goals of these experiments are not well-defined
    - (p. 8) While the authors do define and cite some notions of OOD, subpopulations, and singletons, connections to the main results in Section 3 are not clear.
    - (p. 12) There is a logical leap from the authors’ results to conclusions about proposed MIA efficiency gains. If an adversary *already* has a high quality, efficient, accurate memorization proxy defined as an oracle, then distinguishing between P_0 and P_1 is trivial by the adversary’s construction.

---

> ### Author Response · Authors · 2025-07-24
> **Thanks for the excellent comments! Responses enclosed (part 1)**
>
> **Tautology**: We agree that the extreme case, when a sample is always correctly classified when in the training set and never when out, trivially leads to successful membership inference. However, our contribution lies not in highlighting this boundary case, but in formally characterizing the connection between label memorization and MI advantage in both worst-case and average-case regimes. Our main theorem (Theorem 2) shows that memorization directly lower-bounds the MI advantage under a natural instantiation of the adversary, and our corollary (Corollary 1) connects this memorization score to sample complexity under a hypothesis testing view. These are novel theoretical insights that were not previously established.
>
>
> With respect to RQ2, we fully agree that identifying high-memorization points is nontrivial in practice. Indeed, one of the key insights from our work is that if an adversary has access to even a coarse proxy for memorization (e.g., via external metrics such as TRAK) they can reduce the computational burden of launching MI attacks (e.g., by training fewer shadow models or focusing attention on a small candidate pool). Our theoretical results provide a rigorous justification for this strategy, and our experiments show that this intuition holds across real datasets. We will revise the framing of RQ2 and the “Main Thesis” section to better emphasize that we do not assume access to perfect memorization scores (nor claim to propose a “new practical attack”), but instead investigate how their structure can be exploited when approximate estimates are available.
>
>
> **Prior Literature**: Our intention in Scenario 2 was to adopt the same modeling assumption used in LiRA, where the adversary assumes that the distribution of model outputs (such as logits or prediction scores) follows a parametric form (typically a multivariate Gaussian), conditioned on whether a sample was included in training. That is, the Gaussian assumption is over the distributions $P_0, P_1$ defined on the output space $f(\theta) \in \mathbb{R}^{\ell}$, and not over the distribution of the model parameters $\theta \in \Theta$. While this is consistent with the strategy used in LiRA, we now recognize that our current wording may have inadvertently suggested that the Gaussianity assumption applies to the model (parameter) distribution itself. We will revise the text in Section 4.1 to explicitly clarify that the parametric assumption is made over model outputs (logits or softmax scores), and that this setup mirrors LiRA's empirical strategy. We will also make the estimation procedure in Scenario 2 more precise by emphasizing that shadow models are used to approximate the output distributions under the “in” and “out” hypotheses, which are then used to conduct a soft log-likelihood ratio (sLLR) test.
>
>
> **Theorem 2**: We thank the reviewer for highlighting this concern. Theorem 2 (p.6) establishes that for a specific adversary that checks whether $\theta(x) = y$, the MI advantage is lower-bounded by the memorization score $\texttt{mem}(L, S, z)$. While the conclusion may appear intuitive in hindsight, our goal is to formalize and quantify this connection within the MI security game. This result gives a concrete lower bound on adversary power using a simple, interpretable strategy, and serves as the foundation for our analysis of sample complexity and attack efficiency.
>
>
> The paragraph preceding Equation 7 uses a standard argument from cryptographic game analysis, where we model the adversary as a deterministic algorithm, without loss of generality, by treating its internal randomness explicitly. This lets us reframe MI advantage in terms of expectations over model draws from $L(S)$ and $L(S \cup \{z\})$. We acknowledge that this reasoning was presented too tersely, and we will revise the manuscript to elaborate on the intermediate steps, clearly explain the motivation, and provide additional commentary linking this result to the rest of our framework.

---

> > ### Author Response · Authors · 2025-07-24
> > **Part 2 (final)**
> >
> > **Evaluation**: We appreciate the reviewer’s concern regarding the connection between our theoretical analysis and experimental validation. We agree that the connection could have been made more explicit, and we will revise the introduction to Section 5 and the captions in the figures to emphasize this alignment.
> >
> >
> > Each experimental question corresponds directly to one of the theoretical claims in our theory. Specifically:
> > RQ1 empirically tests Theorem 2 and Corollary 1 by measuring MIA success as a function of sample memorization. These results show that memorization is not merely a post hoc explanation but a measurable predictor of attack advantage.
> > RQ2 examines the implications of our sample complexity bounds by varying the number of shadow models and observing whether highly memorized samples allow for more efficient attacks.
> >
> >
> > Regarding the oracle assumption: we emphasize in the paper that the experiments are not intended to suggest a practical attack pipeline, but to quantify how memorization contributes to success and efficiency in MIAs. The memorization oracle is a tool that lets us explore the upper bound of what an idealized adversary might achieve.
> >
> >
> > We will revise the framing of RQ2 to explicitly state that the experiments are testing how much memorization matters, not how to infer memorization, and that the empirical curves provide actionable insight into the impact of sample choice on MIA overhead.
> >
> > *In conclusion: thanks for the great questions and feedback in helping our paper! We are happy to respond to any further questions/clarifications.*

---

> > > ### Author Response · Authors · 2025-07-28
> > > **Ping!**
> > >
> > > Hello,
> > >
> > > We are happy to answer any further questions you may have. We understand that this period conflicts with the NeurIPS discussion period, but we are happy to engage with any further comments.

---

> > > > ### Author Response · Authors · 2025-08-01
> > > > **Ping x 2!**
> > > >
> > > > Hello,
> > > >
> > > > As we are nearing the end of the discussion period, we are writing to inquire if our responses have answered your questions. We are happy to answer more questions you may have, if you give us sufficient time to respond.
> > > >
> > > > Thanks for your time!

---

> > > > > ### Comment · Reviewer_uCxk · 2025-08-04
> > > > >
> > > > > Thank you for the detailed rebuttal and revision!

---

### Review · Reviewer_YzjB · 2025-07-20

**Summary Of Contributions:**

This paper establishes a formal theoretical connection between label memorization and membership inference attack (MIA) success. The authors prove that a sample's memorization score introduced by Feldman-Zhang 2019 directly lower-bounds an adversary's advantage in determining whether that sample was used in training. This result shows why certain data points are more vulnerable to privacy attacks than others.

Their empirical results demonstrate a strong correlation between memorization scores and MIA vulnerability across multiple experimental settings. The experiments compare MIA performance on three categories of samples: random out-of-distribution (OOD) data, OOD samples from specific subpopulations, and OOD samples with high memorization scores (which they call "singletons").

**Audience:**

Yes

**Claims And Evidence:**

No

**Requested Changes:**

please see above.

**Strengths And Weaknesses:**

In general, I think this paper is not very well-written and many statements are difficult to digest. I also have a major comment about the theoretical result. Their empirical results are quite interesting. However, I should note that the fact that highly memorized samples are easier to attack seems obvious.

More specific comments:

1- The definition of MIA in this paper is rather non-standard. In particular, in MIA it is often assumed that the attacker knows the model and a datapoint. Here, the authors consider a version of MIA which is similar to the definition of DP. The attacker knows the dataset S and a point z, there are two hypotheses, either the true training set is S or S ∪ {z}. This form of MIA is not what the community thinks of as MIA. As a result, I think using terms such as shadow models and comparing with Carlini et al. or Shokri et al. seems not fair. In particular, in Carlini et al. the assumption is that the attacker only knows the model output and the data point. Also, during training of shadow models the attacker uses an independent training set sampled from the same distribution, not the actual training dataset S.

2- Major comment: Equation 8 and Corollary 1 seem not correct to me. In particular, we can start with the definition of TV

TV(P_0 , P_1) = sup_{A} P_0(A) - P_1(A),

where A is a measurable set. Let z=(x,y). Then, by choosing A={θ: θ(x)=y}, we have TV(P_0 , P_1) ≥ mem(z).
Also, we have H^2(P_0 , P_1) ≤ TV(P_0 , P_1). Therefore, you can't claim that H^2(P_0 , P_1) ≤ mem(z).

Other comments:

1- The structure of the introduction section can be improved. Currently, it is not clear the problem statement and contribution. Also, the authors should put their results into the context of prior work.

2- Brown et al. is not related to label memorization. Their measure of memorization is data reconstruction.

3- As mentioned, the security game in the paper is from Mahloujifar et al., 2022. The authors should state the security game that shows the threat model considered in the paper (Section 2.2).

4- The definition of sample complexity seems to have a typo: In the second part where the optimal test is defined, what is error probability? It seems it should be the advantage instead of error probability.

5- In the "our main thesis" section, what is m*? Also, the main thesis seems straightforward. I may miss the interesting aspect of the claim, however.

6- Equation 7 seems to have a typo. Should it be Pr_{θ ∼ L} instead of \in?

7- Corollary 1: It is not the correct usage of \tilde{\Theta}. Often \tilde{\Theta} refers to asymptotic notation that hides logarithmic factors.

8- Lemma 2: "The optimal sample complexity of a test" is not correct terminology. The optimal sample complexity refers to the sample complexity of the optimal test.

---

> ### Author Response · Authors · 2025-07-24
> **Thanks for your excellent feedback! Responses enclosed (part 1)**
>
> **Novelty**: We appreciate the reviewer’s comment, but respectfully disagree with the claim that “the fact that highly memorized samples are easier to attack seems obvious.” While this may appear intuitive in retrospect, the relationship between memorization and membership inference vulnerability has neither been universally assumed nor thoroughly and formally characterized in prior work. Historically, many key insights in machine learning—such as the generalization gap between training and test sets—may seem “obvious,” yet they form the basis of deep theoretical frameworks. The same holds true here: while it might seem natural to expect that memorized points are more vulnerable, the literature shows this relationship is far from trivial. For example, Carlini et al. (2022) demonstrate that test points can also be misclassified as members.
> What is novel in our work is not merely reaffirming this relationship, but precisely characterizing how memorization impacts privacy risk. We provide theoretical results showing that an adversary’s advantage can be explicitly tied to the model’s ability to distinguish between distributions that differ by a single sample, thus grounding the intuition in formal terms. To our knowledge, no prior work establishes this theoretical connection.
>
> **Definitions**: While we agree that our setup differs from the canonical formulation, where the adversary is given access to a trained model and a point $z$, our definition is deliberately chosen to align with the hypothesis testing perspective used in differential privacy (DP), in which the attacker distinguishes between $S$ and $S \cup \{z\}$. This framing enables theoretical analysis of the adversary’s power and allows us to directly quantify privacy leakage in terms of information-theoretic bounds. We respectfully push back on the claim that our setting is incompatible with the terminology or comparisons drawn from prior MIA work, for the following reasons:
>
> - On shadow models: Our usage of “shadow models” is consistent with Shokri et al.’s (2017) original formulation, where shadow models are trained on subsamples of the actual training data, and not only entirely independent draws from the underlying data distribution. In practice, this means that many MIA implementations operate under assumptions similar to ours,where the attacker has partial or approximate access to the training dataset. We will clarify this point in the text, but we believe the terminology is appropriate and historically grounded.
> - On comparison with prior MIA methods: While our adversarial model is stronger—assuming access to $S$—we argue that many MIA implementations implicitly adopt similar levels of access. In Shokri et al. (2017) and follow-up works, the attacker is often given auxiliary data that is derived from the training set itself. Thus, our comparisons are not intended to claim equivalence of assumptions, but rather to illustrate how attack success varies across attacker regimes. We will revise the manuscript to make these distinctions clearer and avoid overstating direct comparability.
> - On theoretical contributions: We acknowledge that prior work such as Ye et al. (2022) includes initial theoretical framing. However, our formulation allows for more explicit information-theoretic analysis, connecting generalization, overfitting, and privacy leakage in a unified way. This level of formalism is absent in most empirical MIA studies, and we view it as a key contribution of our work.
> In summary, our formulation explores a stronger but analytically tractable setting that bridges empirical MIA and formal privacy guarantees. It captures attacker behaviors already implicit in many MIA implementations (e.g., use of subsampled training data), while enabling principled theoretical insight. We will revise the paper to more clearly delineate our assumptions, defend our terminology, and contextualize comparisons to prior work.
>
> In summary, our formulation explores a stronger but analytically tractable setting that bridges empirical MIA and formal privacy guarantees. It captures attacker behaviors already implicit in many MIA implementations (e.g., use of subsampled training data in the *actual* implementation of Carlini et al., 2022a; please refer to https://github.com/tensorflow/privacy/blob/master/research/mi_lira_2021/train.py#L214-L228 from their official implementation), while enabling principled theoretical insight. We will revise the paper to more clearly delineate our assumptions, defend our terminology, and contextualize comparisons to prior work.

---

> > ### Author Response · Authors · 2025-07-24
> > **Response part 2**
> >
> > **Main thesis**: $m^\ast$ denotes the minimal number of shadow models required to achieve a target level of membership inference advantage (e.g., at a fixed true positive rate and low false positive rate), when the adversary targets a sample with perfect memorization score ($\texttt{mem}(L, S, z) = 1$). We will revise the notation and add this clarification to the paper.
> >
> > Regarding the substance of our main thesis: while the claim that highly memorized points are easier to attack may seem intuitive, our contribution lies in formally quantifying and proving this connection. We connect label memorization to (1) the worst-case adversary’s MI advantage (Theorem 2) and (2) the sample complexity required for MI under a hypothesis testing framework (Corollary 1). This formalism enables us to explain disparate MI performance across samples and derive computational implications; specifically, that adversaries can reduce overhead by focusing on high-memorization samples.
> >
> > **Correctness of the theory (Equation 8 and Corollary 1)**:
> > You have raised an excellent and subtle point regarding the justification for Equation 8. We agree entirely with your analysis that the inequality $H^2(P_0, P_1) \leq \texttt{mem}(L, S, z)$ is not generally valid without and additional assumption, and we thank you for identifying this gap in our original argument. Here we clarify the connection and limitations.
> > Our memorization score, $\texttt{mem}(L, S, z)$, is defined as the absolute difference in correctness probabilities:
> > $$
> > \texttt{mem}(L, S, z) = \left| \Pr_{\theta \sim P_1}[\theta(x) = y] - \Pr_{\theta \sim P_0}[\theta(x) = y] \right|,
> > $$
> > where $P_1 = L(S \cup \{z\})$ and $P_0 = L(S)$ are distributions over models. This is a pointwise difference in marginal prediction behavior for the fixed input-label pair $(x, y)$. In contrast, the Hellinger distance $H^2(P_0, P_1)$ is a global divergence between distributions $P_0$ and $P_1$ over the entire model space. It quantifies how distinguishable these two distributions are in totality.
> > To relate the two, we consider the total variation distance:
> > $$
> > \text{TV}(P_0, P_1) = \sup_ {A} \left| \Pr_{P_0}[A] - \Pr_{P_1}[A] \right|.
> > $$
> > By choosing the measurable set $A = \{ \theta : \theta(x) = y \}$, we obtain:
> > $$
> > \text{TV}(P_0, P_1) \geq \texttt{mem}(L, S, z),
> > $$
> > since this choice of $A$ captures exactly the quantity defined as memorization. Additionally, the well-known inequality $H^2(P_0, P_1) \leq \text{TV}(P_0, P_1)$ allows us to write:
> > $$
> > H^2(P_0, P_1) \leq \text{TV}(P_0, P_1) \geq \texttt{mem}(L, S, z).
> > $$
> > Thus, unless the set $A = \{ \theta : \theta(x) = y \}$ achieves (or nearly achieves) the supremum in the definition of total variation, the inequality $H^2(P_0, P_1) \leq \texttt{mem}(L, S, z)$ is not generally valid. Our earlier presentation *implicitly assumed* that $A$ is a dominant contributor to $\text{TV}(P_0, P_1)$. We recognize that this implicit condition represents an idealized version of the core observation that underpins many classic MIAs: the idea that a model’s behavior changes most significantly on the specific examples it was trained on. While the true TV distance $\text{TV}(P_0, P_1) $ certainly captures global changes to the model, the success of many MIAs relies on the signal from $mem(z)$ being particularly strong and distinct.
> >
> >
> > To formally bridge the gap in our proof and make this connection explicit, we will revise the manuscript to introduce the following *explicit modeling assumption* to clarify the conditions under which the inequality is valid:
> >
> > *Prediction-Aligned Divergence Assumption*: For our theoretical analysis, we assume a setting where the dominant change between the model distributions $P_0$ and $P_1$ arises from the shift in the model’s predicted label on input z. Formally, we analyze the case where the measurable set $A = \{f(\theta) : \theta(x) = y\}$ is the primary contributor to the mass difference, such that $\text{TV}(P_0, P_1) = |mem(z)| = mem(z)$ (for most practical algorithms we expect this value to be non-negative; Section 4.1 in Feldman et al., (2020))
> >
> > We will clarify that while this is an idealized condition, it allows for a tractable analysis that isolates the very mechanism many MIAs are designed to exploit. We believe this is a reasonable lens for theoretical examination, and our empirical findings, which demonstrate a strong link between high memorization scores and MIA success, support the practical relevance of focusing on this specific effect.
> > This clarification adds important nuance and precision to our theoretical claims. Thank you again for helping us refine this connection and significantly improve the manuscript.

---

> > > ### Author Response · Authors · 2025-07-24
> > > **Part 3 (final)**
> > >
> > > **Others**
> > > We appreciate the reviewer for the detailed comments. We will revise the manuscript to address them as follows.
> > > - We agree that Brown et al.'s notion of memorization focuses on data reconstruction and is not directly related to label-level memorization as we study. We will remove this citation in contexts discussing prediction-based memorization and replace it with more appropriate references (e.g., Feldman and Zhang, 2020).
> > > - We appreciate the reviewer catching the typo in Equation 7. We will update the notations to $\theta \sim L(S)$ and $\theta \sim L(S \cup \{z\})$, replacing the previously used $\in$ symbol.
> > > - As the reviewer suggested, we will update Corollary 1 to use a notation other than $\tilde{\Theta}$ to avoid confusion.
> > > - The confusion stems from the incorrect reference to error probability $1 - \alpha$ when the definition is in terms of advantage $\alpha$. We have revised the sentence to clarify that sample complexity refers to the number of samples required for a hypothesis test to achieve advantage at least $\alpha$, as defined in Equation 3, and not to a specific error probability.
> > > - We will revise Lemma 2 to state: The sample complexity of the optimal test $T^\star$ to distinguish $P_0$ from $P_1$ is $\Theta(1 / H^2(P_0, P_1))$.
> > >
> > > **References:**
> > >
> > >
> > > - Carlini, Nicholas, et al. "Membership inference attacks from first principles." 2022 IEEE symposium on security and privacy (SP). IEEE, 2022.
> > > - Shokri, Reza, et al. "Membership inference attacks against machine learning models." 2017 IEEE symposium on security and privacy (SP). IEEE, 2017.
> > > - Feldman, Vitaly, and Chiyuan Zhang. "What neural networks memorize and why: Discovering the long tail via influence estimation." Advances in Neural Information Processing Systems 33 (2020): 2881-2891.
> > > - Feldman, Vitaly. "Does learning require memorization? a short tale about a long tail." Proceedings of the 52nd annual ACM SIGACT symposium on theory of computing. 2020.
> > > - Ye, Jiayuan, et al. "Enhanced membership inference attacks against machine learning models." Proceedings of the 2022 ACM SIGSAC conference on computer and communications security. 2022.
> > >
> > > *In conclusion: thanks for the excellent feedback! We are happy to answer any further questions/clarifications you may have.*

---

> > > > ### Author Response · Authors · 2025-07-28
> > > > **Ping!**
> > > >
> > > > Hello,
> > > >
> > > > We are happy to answer any further questions you may have. We understand that this period conflicts with the NeurIPS discussion period, but we are happy to engage with any further comments.

---

> > > > > ### Author Response · Authors · 2025-08-01
> > > > > **Ping x 2!**
> > > > >
> > > > > Hello,
> > > > >
> > > > > As we are nearing the end of the discussion period, we are writing to inquire if our responses have answered your questions. We are happy to answer more questions you may have, if you give us sufficient time to respond.
> > > > >
> > > > > Thanks for your time!

---

> > > > > > ### Comment · Reviewer_YzjB · 2025-08-01
> > > > > >
> > > > > > Thanks for your response. I don't have any additional questions.

---

> > > > > > > ### Author Response · Authors · 2025-08-01
> > > > > > > **Thanks for your feedback!**
> > > > > > >
> > > > > > > Thank you for the great questions and feedback in helping refine our submission. We appreciate your time and effort!

---

### Author Response · Authors · 2025-08-12
**Summary of Revisions**

We thank all the reviewers for their constructive and valuable feedback, which has helped us improve our paper. We have submitted our revised manuscript to address your comments, including the following updates:

- **Novelty and Contributions:** Updated the introduction to clarify that the novelty lies in formalizing the link between memorization and MIA advantage/sample complexity via information-theoretic and hypothesis-testing tools, and that this is the first formal lower bound on MI advantage using per-sample memorization (Reviewer `YzjB`).

- **Corrected Main Theoretical Results**: revised Equation 7 (which was Equation 8 in the previous manuscript) with a proof sketch, along with a discussion on the additional Prediction-Aligned Divergence Assumption and its limitations in Appendix B.1. Corrected Corollary 1 to clarify them as a lower bound, and note empirical tightness (Reviewer `YzjB` and `MYbu`).

- **Connections to Differential Privacy (DP):** expanded the discussion in Appendix E and the threat model in Section 2.2, linking memorization to per-sample privacy leakage in DP, and distinguishing instance-level leakage from worst-case DP bounds (Reviewer `MYbu`).

- **Practicality and Experimental Framing:** Clarified that experiments test theory rather than propose a deployable attack, and that the memorization oracle is idealized. In Section 4, reframed RQ2 to focus on approximate proxies (e.g., TRAK) and tied each experiment to the relevant theoretical claim in captions (Reviewer `uCxk`).

- **Definitions and Assumptions:** Explicitly stated that the MI setting assumes access to $S$ (DP-style $S$ vs. $S \cup \{z\}$). Defined "shadow models" consistent with Shokri et al. (2017) and clarified "memorization" as label-level memorization (removing reconstruction references) (Reviewer `YzjB`).

Revisions are highlighted in blue in the text for easy identification. If there are any remaining suggested changes, we look forward to incorporating them into the final version of our paper or having further discussions with the reviewers as needed.

Thank you,\
The Authors

---

### Decision · Action_Editor_XqET · 2025-09-18

**Recommendation:** Reject

**Audience:**

Yes

**Audience Explanation:**

The topic is clearly relevant and builds on an established line of work of interest to the TMLR audience.

**Claims And Evidence:**

No

**Claims Explanation:**

Multiple reviewers noted one or more issues with
1. Incorrect proofs of important claims, or
2. Experiments that don't support claims.

**Resubmission Of Major Revision:**

The authors may consider submitting a major revision at a later time.